# The operational cloud retrieval algorithms from TROPOMI on board Sentinel-5 Precursor

Diego G. Loyola[1], Sebastián Gimeno García[1], Ronny Lutz[1], Athina Argyrouli[3,1], Fabian Romahn[1], Robert J.D. Spurr[2], Mattia Pedergnana[1], Adrian Doicu[1], Victor Molina Garcia[1], Olena Schüssler[1]

[1]German Aerospace Centre (DLR), Remote Sensing Technology Institute, Oberpfaffenhofen, 82234 Wessling, Germany
[2]RT Solutions, Inc., 9 Channing Street, Cambridge, MA 02138, USA
[3]Technical University of Munich, TUM Department of Civil, Geo and Environmental Engineering, Chair of Remote Sensing Technology, Germany

*Correspondence to*: Diego Loyola (Diego.Loyola@dlr.de)

**Abstract.** This paper presents the operational cloud retrieval algorithms for the TROPOspheric Monitoring Instrument (TROPOMI) on board the European Space Agency Sentinel-5 Precursor (S5P) mission scheduled for launch in 2017.

Two algorithms working in tandem are used for retrieving cloud properties: OCRA (Optical Cloud Recognition Algorithm) and ROCINN (Retrieval of Cloud Information using Neural Networks). OCRA retrieves the cloud fraction using TROPOMI measurements in the UV/VIS spectral regions and ROCINN retrieves the cloud top height (pressure) and optical thickness (albedo) using TROPOMI measurements in and around the oxygen *A*-band in the NIR.

Cloud parameters from TROPOMI/S5P will be used not only for enhancing the accuracy of trace gas retrievals, but also for extending the satellite data record of cloud information derived from oxygen *A*-band measurements, a record initiated with GOME/ERS-2 over twenty years ago.

The OCRA and ROCINN algorithms are integrated in the S5P operational processor UPAS (Universal Processor for UV/VIS/NIR Atmospheric Spectrometers), and we present here UPAS cloud results using OMI and GOME-2 measurements. In addition, we examine anticipated challenges for the TROPOMI/S5P cloud retrieval algorithms and we discuss the future validation needs for OCRA and ROCINN.

## 1. Introduction

Clouds are an important component of the hydrological cycle and play a major role in the Earth's climate system through their strong impact on radiation processes. The interplay of sunlight with clouds imposes major challenges for satellite remote sensing, both in terms of the spatial complexity of real clouds and the dominance of multiple scattering in radiation transport. The retrieval of trace gas products from TROPOMI/S5P will be strongly affected by the presence of clouds.

The physics behind the influence of cloud on trace gas retrieval is well understood, and in general, there are three different contributions (Liu et al., 2004; Kokhanovsky and Rozanov, 2008; Stammes et al., 2008; Wagner et al., 2008): (a) the albedo

effect associated with the enhancement of reflectivity for cloudy scenes compared to cloud-free sky scenes, (b) the so-called shielding effect, by which that part of the trace gas column below the cloud is hidden by the clouds themselves, and (c) the increase in absorption within the cloud, related to intra-cloud multiple scattering enhancements of optical path lengths. The albedo and in-cloud absorption effects increase the visibility of trace gases at and above the cloud top, while the shielding effect (if not corrected for) normally results in an underestimation of the trace gas column.

Using radiative transfer modelling, several papers have quantified the influence of cloud parameters on the retrieval of trace gas columns (Liu et al., 2004; Ahmad et al., 2004; Boersma et al., 2004; Van Roozendael et al., 2006; Kokhanovsky et al., 2007; du Piesanie et al., 2013, Doicu et al., 2014). These studies have shown that cloud fraction, cloud optical thickness (albedo), and cloud top pressure (height) are the most important quantities determining cloud correction of satellite trace gas retrievals.

Use of the oxygen $A$-band in the NIR generates complementary cloud information (especially for low clouds), as compared to traditional thermal infrared (TIR) sensors (as used in most meteorological satellites) that are less sensitive to low clouds due to reduced thermal contrast. Recent studies on the NIR-TIR comparison (Stengel et al., 2017) demonstrate that indeed a critical improvement appears in the cloud top height retrieval when the $O_2$ $A$-band is used. Lelli et al., (2016) found an underestimation of roughly 0.6 - 1.0 km when retrieving low clouds only in the TIR.

TROPOMI (Veefkind et al., 2012) has eight spectral bands covering the UV, VIS, NIR, and SWIR spectral regions, and an unprecedented spatial resolution of 7x3.5 km² at nadir for bands 2-6, from which measurements most of the trace gas, aerosol and cloud properties will be retrieved. Another band in the UV (band 1) has a resolution of 28x7 km$^2$ at nadir and bands 7-8 in the SWIR (where the green-house gases are retrieved) have spatial resolution 7x7 km$^2$ at nadir. TROPOMI will fly on board S5P in a sun-synchronous polar orbit providing a daily global coverage with a wide swath of 2600 km and a local overpass time of 13:00 at the ascending node. TROPOMI/S5P will be the first atmospheric composition mission of the European Copernicus programme and TROPOMI's 7-year lifetime will extend the unique data record started more than 20 years ago with GOME/ERS-2, SCIAMACHY/ENVISAT, OMI/AURA, GOME-2/MetOp-A and GOME-2/MetOp-B, which have local overpass times of 10:30 (descending node), 10:00 (descending node), 13:30 (ascending node), 09:30 (descending node) and 08:45 (descending node), respectively.

This paper provides a detailed description of the operational TROPOMI/S5P cloud retrieval algorithms. We start with a short overview in Section 2. In Sections 3 and 4, we present the OCRA algorithm for the cloud fraction retrieval using TROPOMI measurements in the UV/VIS spectral regions and the ROCINN algorithm for the retrieval of cloud top height (pressure) and optical thickness (albedo) using TROPOMI measurements in and around the oxygen $A$-band in the NIR. The error budget of the OCRA and ROCINN retrievals is described in Section 5 and results from application of the S5P algorithms to OMI and GOME-2 measurements are presented in Section 6. In Section 7, we discuss anticipated challenges for the TROPOMI/S5P cloud retrieval algorithms, and the future validation needs for OCRA and ROCINN.

## 2. Overview of the cloud retrieval algorithms

The operational TROPOMI/S5P cloud properties are retrieved using two algorithms working in tandem: OCRA and ROCINN.

OCRA derives the cloud fraction from UV/VIS radiances by separating the sensor measurements into two components: a cloud-free background and a remainder expressing the influence of clouds. OCRA was first developed for GOME/ERS-2 in the late 1990s using data from GOME's broad-band PMDs (Polarization Measurement Devices). OCRA has also been applied operationally to SCIAMACHY and GOME-2. Initial cloud-free backgrounds for these sensors were based on GOME data before dedicated measurements became available from SCIAMACHY and GOME-2. In this paper we present the adaptation of OCRA to TROPOMI/S5P using UV/VIS radiances themselves (instead of PMD measurements), with an initial cloud-free background based on OMI data.

ROCINN is based on the comparison of measured and simulated satellite sun-normalized radiances in and near the $O_2$ $A$-band to retrieve cloud height and cloud optical thickness. ROCINN uses the cloud fraction from OCRA as an input. Two sets of TROPOMI/S5P cloud properties will be provided by ROCINN: (a) cloud top height and cloud top albedo using the "Clouds-as-Reflecting-Boundaries" (CRB) model in which clouds are treated as simple Lambertian surfaces; and (b) cloud top height and cloud optical thickness using a more realistic "Clouds-As-Layers" (CAL) model in which clouds are treated as optically uniform layers of light-scattering particles (water droplets).

OCRA and ROCINN are being used for the operational retrieval of trace gases from GOME (Van Roozendael et al., 2006), GOME-2 (Loyola et al., 2011; Valks et al., 2011; Hao et al., 2014). In a similar manner, OCRA and ROCINN results will be used in the following operational TROPOMI/S5P trace gas retrieval products: total ozone (Loyola et al., 2017), tropospheric ozone (Heue et al., 2016), formaldehyde, and sulfur dioxide (Theys et al., 2017).

In this paper we present for the first time the latest developments of the ROCINN algorithm (incorporating both CAL and CRB models). CAL is the preferred method for the relatively small TROPOMI/S5P ground pixels (7x3.5 km$^2$). The CRB approach works best with large pixels (Kokhanovsky et al., 2007) such as those from GOME (footprint 320x40 km$^2$), where different types of clouds are combined and errors on the cloud model are usually self-compensating. Furthermore CAL is more accurate than CRB for optically thin clouds (Rozanov and Kokhanovsky, 2004) and these kinds of clouds are the most frequent on a global scale. Previous studies using TOMS and GOME/SCIAMACHY measurements have demonstrated that a plane-parallel scattering cloud model is superior to a Lambertian reflectance cloud model for trace gas retrievals (Ahmad et al., 2004) and (Diedenhoven et al., 2007) respectively. Furthermore, errors on retrieved $NO_2$ columns can be significantly reduced using cloud parameters from combined UV/VIS and NIR spectral (van Deelen et al., 2008) as obtained from OCRA and ROCINN_CAL. More recent studies have shown that for the smaller GOME-2 pixels, CAL retrieval produces more reliable cloud information than that from CRB (Sihler et al., 2015), not only with regard to the accuracy of the cloud

parameters themselves, but also with respect to the effect of cloud parameter uncertainties on total ozone accuracy (Loyola et al., 2017).

It is important to note that a cloud model similar to CAL is being used for the retrieval of aerosol properties from UV measurements (Torres et al., 2011) in order to overcome the systematic biases induced by using a Lambertian cloud model.

Similarly, it was shown that a plane-parallel scattering cloud model is more appropriate for the estimation of the surface UV irradiance than is the case for a Lambertian reflector cloud model (Krotkov et al., 2001), and this more realistic cloud scattering model will be used for retrieving the UV irradiance from TROPOMI/S5P (Lindfors et al., 2017).

The following subsection gives a short summary of the heritage algorithms used for retrieving cloud information from UV/VIS/NIR spectrometers.

## 2.1. Heritage algorithms

Several cloud retrieval algorithms based on measurements in and around the $O_2$ $A$-band at 760 nm have been developed for the GOME-type of sensors: these include the ICFA (Initial Cloud Fitting Algorithm) (Kuze and Chance, 1994), FRESCO (Fast REtrieval Scheme for Clouds from the Oxygen $A$-band) (Koelemeijer et al., 2001, Wang et al., 2008), SACURA (Semi-Analytical CloUd Retrieval Algorithm) (Rozanov and Kohanovsky, 2004), UV/NIR (Diedenhoven et al., 2007), and

ROCINN algorithms. These are all based on the Independent Pixel Approximation (IPA), which is the assumption that the "radiative properties of a single satellite 'pixel' are considered in isolation from neighbouring pixels" (definition of the American Meteorological Society). The IPA allows for the application of one-dimensional plane-parallel radiative transfer (RT) theory in the forward simulation of cloud-contaminated atmospheric scenarios. This is further discussed in Section 5.

The ICFA algorithm was used in the initial GOME data processing to derive the effective fractional cover. The FRESCO

algorithm, also developed for GOME, is based on the calculation of transmittances (later, single scattering radiances) and it retrieves effective cloud fraction and cloud top pressure, assuming a fixed cloud albedo of 0.8. The SACURA algorithm was developed initially for the SCIAMACHY instrument and then modified to handle also GOME measurements (Lelli et al., 2012). SACURA uses semi-empirical formulae from asymptotic radiative transfer theory to retrieve cloud optical thickness, cloud top height, liquid water path and other parameters. The UV/NIR algorithm uses information from 350 to 390 nm

together with $O_2$ $A$-band to retrieve cloud fraction, cloud optical thickness, and cloud top pressure. The ROCINN algorithm (Loyola et al., 2007) is currently being used in the operational GOME and GOME-2 products and it retrieves as primary quantities the cloud top height and cloud albedo.

The broad-band polarization measurements from GOME, SCIAMACHY and GOME-2 are used for computing cloud fraction; see for example OCRA (Loyola et al., 1998; Lutz et al., 2016) and HICRU (Grzegorski et al., 2006). Enhancements

to these algorithms have been introduced in recent years - see for example the detection of sun glint effects (Loyola et al., 2011; Lutz et al., 2016). For these instruments, the polarization measurement devices (PMD) enable the cloud fraction to be

retrieved at eight times higher spatial resolution (10x40 km$^2$) compared to that for the main science channels (80x40 km$^2$) which are used for the retrieval of the trace gases.

There are three cloud-property algorithms in operational use for the OMI instrument (OMI has no $O_2$ $A$-band measurements). The first (OMCLDRR) uses the cloud screening effect on Fraunhofer filling signatures (due to inelastic rotational Raman scattering) in the region 346-354 nm to derive effective cloud fraction and cloud optical centroid pressure (Joiner and Vasilkov, 2006; Vasilkov et al., 2008; Joiner et al., 2012). This algorithm is based on the mixed Lambertian equivalent reflectivity (MLER) assumption. The second algorithm (OMCLDO2) uses reflectances in and around the $O_2$-$O_2$ absorption band near 477 nm (Acarreta et al., 2004, Veefkind et al., 2016); DOAS-retrieved $O_2$-$O_2$ slant columns are compared with simulated look-up table entries to obtain effective cloud fraction and cloud pressure. The third algorithm (OMAERUV) derives aerosol optical depth and single scattering albedo (Torres et al., 2007) from radiances at 354 and 388nm; the cloud fraction is computed as an intermediate step.

## 3. OCRA

The OCRA cloud fraction determination is based on the comparison between cloud-contaminated measurements and corresponding measurements for the background (cloud-free) surface. A flow chart of the OCRA algorithm is given in Figure 1, and the algorithm steps are described in the following subsections.

A description of the OCRA algorithm and its application to GOME and GOME-2 data is given in (Loyola, 1998; Lutz et al., 2016). For the TROPOMI/S5P application, the new algorithm developments for OCRA are the adaptation to work with 2-colour radiances (GB) using the UV/VIS spectra instead of the 3-colour PMD measurements (RGB) in the UV/VIS/NIR region. The reasons for moving from RGB to GB are twofold: First, the TROPOMI UV/VIS and NIR footprints are spatially mis-aligned, which means that the GB and R colours do not see the exact same footprint, and any misalignment correction would then act as an additional error source in OCRA (see section 5.1). Second, and more importantly, the OMI sensor, which is needed to provide the initial cloud-free reflectance background maps, does not have channels in the red part of the visible spectrum; thus, OMI cannot be used to define a third colour R. These two considerations dictate the need for a two colour approach.

### 3.1. GB-colour conversion

The OCRA colour-space approach can be applied with three colours (RGB space) or two colours (GB space or RG space). For a given location $(x, y)$, we define the reflectance $\rho(x, y, \lambda_i)$ at wavelength range $\lambda_i$ for the footprint of the measurement as:

$$\rho(x, y, \lambda_i) = \frac{\pi \cdot I(\lambda_i)}{E_0(\lambda_i) \cdot cos\theta_0},$$

$\qquad(1)$

where $I(\lambda_i)$ and $E_0(\lambda_i)$ denote the measured earthshine backscattered radiance and the solar irradiance respectively, and $\theta_0$ is the solar zenith angle.

The reflectances used in this algorithm are derived from broad-band measurements of backscattered radiance and extra-terrestrial solar irradiance covering the spectral range of the Green-Blue (GB) colour system. The OCRA spectral ranges with TROPOMI/S5P are 405-495 nm for G and 350-395 nm for B.

### 3.2. OCRA cloud-free background

The core of the algorithm is the construction of a cloud-free composite of multi-temporal (time series of measurements over the same location) reflectances that is independent of atmosphere and solar and viewing angles; this is indicated as the "Internal Store" in the flowchart of Figure 1. For the off-line creation of cloud-free reflectance composites in the GB case, the GB reflectances are translated into normalized $gb$-colour space via the relations

$$g = \frac{\rho(x,y,\lambda_G)}{\sum_{i=GB}\rho(x,y,\lambda_i)}, b = \frac{\rho(x,y,\lambda_B)}{\sum_{i=GB}\rho(x,y,\lambda_i)}, \tag{2}$$

If M is the set of $n$ normalized multi-temporal measurements over the same location $(x,y)$, then a cloud-free (or minimum cloudiness) pixel $gb_{CF} \in$ M is selected using the brightness criterion $\|gb_{CF} - W\| \geq \|gb_k - W\|$, for $k = 1, \cdots n$, where $W$ is the *white point* (1/2, 1/2) in the $gb$-chromaticity diagram. This point refers to a situation where B and G are equal, i.e. there is no wavelength dependence across the UV/VIS region - this is interpreted as a scene fully covered by cloud. Measurements under cloudy conditions are projected to the white point, and the measurement that is most distant from $W$ is considered to be cloud-free. A cloud-free background, labelled "Climatology" in Figure 1, is constructed by merging cloud-free reflectances $\rho_{CF}(\lambda_i)$ (corresponding to $gb_{CF}$) at all locations. It should be noted here that the G and B cloud-free reflectances for a given grid cell are not determined independently as a minimum available reflectance over the whole monthly time range, but rather they are the inter-dependent reflectances belonging to an individual scene representing the largest distance from the white point in the gb-chromaticity diagram.

At the beginning of the TROPOMI/S5P mission, a monthly cloud-free background data set based on OMI measurements will be used, to be replaced by TROPOMI data as the mission unfolds.

### 3.3. Cloud fraction derivation

The radiometric cloud fraction $f_c$ is determined by examining separations between measured GB reflectances and their corresponding cloud-free composite values:

$$f_c = min\left\{1, \sqrt{\sum_{i=GB} \alpha(\lambda_i)\max\{0, [\rho(\lambda_i) - \rho_{CF}(\lambda_i) - \beta(\lambda_i)]\}^2}\right\}. \tag{3}$$

This equation expresses the distance between actual measurements and the corresponding cloud-free scene in colour-space. Scaling factors $\alpha(\lambda_{i=GB})$ define the upper limit for reflectances under fully cloudy conditions, while offsets $\beta(\lambda_{i=GB})$

account for aerosol and other radiative effects in the atmosphere and as a lower limit basically define the cloud free conditions. The $max\{\}$ and $min\{\}$ functions ensure that the cloud fraction is confined to the interval [0, 1].

The scaling and offset factors are determined off-line using representative daily global satellite measurements (Lutz et al., 2016). The offsets are the histogram modes from the differences $\{\rho(\lambda_i) - \rho_{CF}(\lambda_i)\}$ and the scaling factors are the inverses of the 99[th] percentile of the cumulative histograms from the differences $\{\rho(\lambda_i) - \rho_{CF}(\lambda_i)\}^2$. The temporal variability of the offsets and scaling factors is investigated by comparing several daily global histograms at different occasions throughout the year. Since no significant seasonal dependence is apparent, only one set of α and β per colour is used. It should be noted here that the offsets may partially compensate for extremely dark scenes (e.g. shadows) and radiative effects (e.g. absorbing aerosols) but a strict separation of aerosols and clouds is not done by OCRA.

## 3.4. Sun glint flagging

Direct sunlight reflected by the ocean surface may reach the satellite sensor, enhancing the measured signal in a manner which contaminates cloud effects. Sun-glint flagging was developed as a component of the operational OCRA algorithm (Loyola, 2011) for GOME-2/MetOp-A, and this treatment was further enhanced using the polarization Stokes fractions (Lutz et al., 2016) to correct for the sun glint effect. Since TROPOMI does not provide polarization information, a simplified sun glint flagging procedure will be used instead of this correction. First, those areas that might be affected by sun glint are marked using the viewing geometry conditions of the measurement:

$$v = \sqrt{(|\Theta_o - \Theta| - 2)^2 + (\varphi_o - \varphi - 180)^2} \, , \qquad\qquad (4)$$

where $\Theta_o, \Theta$ are the solar and satellite zenith angles respectively, and $\varphi_o, \varphi$ the solar and satellite azimuth angles (values are given in degrees). A sun glint flag is set whenever the value of "marker" $v$ is larger than a given threshold. This threshold value will be determined dynamically when real TROPOMI data become available; for now, the OMI threshold value of 45 will be used initially.

## 4. ROCINN

ROCINN is based on the comparison of measured and simulated radiances in and near the O$_2$ $A$-band for retrieving the cloud optical thickness, cloud height and albedo. A flow chart of the ROCINN algorithm is given in Figure 2; the algorithm steps are described in the following subsections.

Previous versions of the ROCINN algorithm for operational processing of GOME (Van Roozendael et al., 2006) and GOME-2 (Loyola et al., 2011) modelled clouds as simple Lambertian surfaces (ROCINN_CRB). The CRB approach was originally developed for GOME (footprint 320x40 km$^2$), where different types of clouds are combined in the large satellite pixels and errors in the cloud model are compensated (Kokhanovsky et al., 2007), but the limitations of the CRB model are

already noticeable with GOME-2 (footprint 80/40x40 km$^2$) where an intra-cloud correction was developed to compensate the CRB overestimation of the $O_3$ ghost column (Loyola et al., 2011). For TROPOMI/S5P, with significantly smaller ground pixels (footprint 7x3.5 km$^2$), we have developed the more sophisticated ROCINN_CAL algorithm presented in this paper. In CAL, clouds are modelled as optically uniform layers of scattering water droplets – with this more physically realistic

scenario, CAL is expected to be more accurate than CRB, especially for optically thin clouds (Rozanov and Kokhanovsky, 2004). For implementation of CAL model, a detailed description on the parameterization of liquid water clouds in the forward model is provided in Section 4.4.

Another change from older ROCINN versions is with the use of the neural networks. In previous ROCINN versions, a neural network was used for solving the inverse function (Loyola et al., 2007) whereas in this version, a neural network is used for

parameterizing the forward model (section 4.4) while the inversion is performed using Tikhonov regularization (section 4.5). This change in methodology enables us to conduct proper error characterizations for all retrievals (section 4.6).

### 4.1. Wavelength recalibration

Before we describe ROCINN itself, we remark on the initial wavelength registration (see Figure 2). The wavelength grid of the measured solar irradiance $E_0$ is recalibrated using a high-resolution solar reference $E_{sol}$ by first dividing the fitting

window into sub-windows and computing for each sub-window $j$ a wavelength shift $\Delta\lambda_j$ between $E_{0,j}$ and $E_{sol,j}$. The solar reference spectrum used for the wavelength calibration is the SAO2010 (https://www.cfa.harvard.edu/atmosphere/links/sao2010.solref.converted) produced for atmospheric measurements in the UV-VIS-NIR by Chance and Kurucz, (2010). The calibration is applied on the reference spectrum by using Differential Optical Absorption Spectroscopy (DOAS) fit methods as part of the UPAS processor. The recalibrated grid is then

established by applying (at each original wavelength point) a shift value computed from a polynomial fit through the $\Delta\lambda_j$ for the various sub-windows. The fitting is achieved for polynomials of a degree of 3 for S5P.

Note that during the inversion (see section 4.5) a wavelength shift for the earthshine spectrum is fitted additionally.

The sun-normalized radiance $R(\lambda)$ at wavelength $\lambda$ is then defined as:

$$R(\lambda) = \frac{I(\lambda)}{E_0(\lambda)},$$

(5)

where $I(\lambda)$ and $E_0(\lambda)$ denote the measured earthshine backscattered radiance and solar irradiance spectra respectively, both spectra registered on the recalibrated solar irradiance grid as noted above.

### 4.2. ROCINN_CAL

For ROCINN with CAL (Clouds-As-Layers), the total sun-normalized radiance is taken to be a linearly-weighted sum of independent radiances $R_s$ for the clear-sky scene and $R_c^{CAL}$ for the cloud-filled scene, with the weighting expressed through

the radiometric cloud fraction $f_c$. Both radiance contributions are calculated using standard one-dimensional radiative transfer models.

The sun-normalized radiance for a cloudy scene is calculated with the cloud treated as a single scattering layer with geometrical extent characterized by cloud top height $Z_{ct}$ and cloud base height $Z_{cb}$ (or alternatively the cloud geometrical thickness $H_c = Z_{ct} - Z_{cb}$). The entire cloud is optically uniform with cloud optical thickness $\tau_c$ and its scattering properties are determined through Mie-scattering calculations for water droplet particles (microphysical properties are discussed below). In the IPA, we may write sun-normalized CAL simulated radiances $R_{sim}^{CAL}$ as:

$$R_{sim}^{CAL}(\lambda) = f_c R_c^{CAL}(\lambda, \Theta, \tau_c, Z_{ct}, Z_{cb}, A_s, Z_s) + (1 - f_c)R_s(\lambda, \Theta, A_s, Z_s) \ . \tag{6}$$

Here, $\Theta$ denotes path geometry (solar and line-of-sight angles), and surface properties are the Lambertian albedo $A_s$ and lower boundary height $Z_s$.

Radiances for clear-sky and cloudy scenarios are calculated using the VLIDORT radiative transfer (RT) code (Spurr, 2006), at wavelengths in and adjacent to the $O_2$ $A$-band. Details of the radiative transfer model (RTM) calculations are given in section 4.4 below.

A complete data set of simulated sun-normalized radiance templates is created off-line for an appropriate range of viewing/solar geometries and surface geophysical scenarios, and for various combinations of cloud properties.

The inverse problem uses least-squares fitting with a generalized form of Tikhonov regularization (details in section 4.5). Retrieval in the $O_2$ $A$-band with the 4-element state vector $\{\tau_c, Z_{ct}, Z_{cb}, f_c\}$ is an ill-posed problem that requires additional information in order to obtain an inverse solution, as there are only two degrees-of-freedom-of-signal (Schüssler et al., 2014). For ROCINN[CAL], the retrieval state vector is just $\{\tau_c, Z_{ct}\}$ for cloud optical thickness $\tau_c$ and height $Z_{ct}$, a fixed cloud geometrical thickness of one kilometre is assumed and the radiometric cloud fraction $f_c$ is taken from OCRA.

### 4.3. ROCINN_CRB

ROCINN with CRB (Clouds-as-Reflecting-Boundary) assumes that clouds are treated as Lambertian reflectors. The sun-normalized CRB simulated radiances $R_{sim}^{CRB}$ are defined as:

$$R_{sim}^{CRB}(\lambda) = f_c R_c(\lambda, \Theta, A_c, Z_c) + (1 - f_c)R_s(\lambda, \Theta, A_s, Z_s) \ . \tag{7}$$

The retrieval state vector for ROCINN[CRB] is $\{A_c, Z_c\}$ for cloud albedo $A_c$ and cloud height $Z_c$; the radiometric cloud fraction $f_c$ is again from OCRA.

### 4.4. Forward model

ROCINN is based on simulated sun-normalized radiances at wavelengths in and around the $O_2$ $A$-band. Two sets of radiance templates were calculated using the CRB and CAL models. The cloudy-scene sun-normalized radiances $R_c^{CAL}$ were

calculated for a multi-layer atmosphere including multiple-scattering in all layers. Mie scattering was used to generate cloud optical properties. Details may be found in (Schüssler et al., 2014).

Simulated sun-normalized radiances $R_{sim}(\lambda)$ are calculated using the vector VLIDORT multiple scattering multi-layer discrete ordinate RTM (Spurr, 2006); the desired total intensity $I$ will incorporate the effects of polarization. The

incorporation of a vector RTM is necessary not for TROPOMI itself but for the processing of data from GOME, SCIAMACHY and GOME-2. In addition to the cloud layers, VLIDORT calculations are based on clear sky optical properties for line absorption by oxygen molecules and Rayleigh scattering by air molecules.

For the line absorption, it is necessary to calculate line-by-line (LBL) radiances (typically at resolution 0.0015 nm for the range 758-771 nm) using line-spectroscopic information for the O$_2$ $A$-band, before convolution with the sensor slit function.

The spectroscopic data is taken from the HITRAN 2012 database (released in June 2013). Absorption cross-sections are computed using LBL software from DLR (Schreier and Schimpf, 2001; Schreier, 2011), in which line absorption signatures are accurately modelled with the Voigt profile.

For Mie scattering calculations, we require knowledge of microphysical properties of clouds consisting of liquid water. The droplets are assumed to be randomly distributed within the cloud layer and any possible in-homogeneity in the cloud is

assumed negligible in the current version of CAL model. In addition, we have found that the consistency of cloud models (e.g. CAL or CRB) used in both the cloud and UV/VIS trace gas retrievals is far more critical than the optical properties selected for the RTM simulations of the CAL templates.

The drop size distribution is well approximated by the modified-Gamma size distribution function (Deirmendjian, 1964):

$$n(r) = Cr^{-\alpha}\exp\left[-\frac{\alpha}{\gamma}\left(\frac{r}{r_c}\right)^{\gamma}\right], \tag{8}$$

which is parameterized by the mode radius $r_c$ in [$um$] and constants α and γ describing the shape of the distribution following (Hess et al., 1998). In Eqn. (8), $C$ is the normalization constant. Characteristic values for the low-level cloud (i.e., Stratus/Cumulus) parameterization are mode radius $r_c = 4.75\ \mu m$ and shape parameters $\alpha = 5$ and $\gamma = 1.61$.

The cloud *macro-physical* properties (classifications of cloud top height and cloud geometrical thickness) are based on the tables in (Wang et al., 2000). Details of this algorithm prototype may be found in (Schüssler et al., 2014).

The cloud geometrical thickness is always constant and equal to 1 km and thus, the liquid water path of the cloud is then defined by the total number concentration. A single phase scattering function is used with the extinction cross section for spherical particles obtained by Mie theory (Van de Hulst, 1957; Bohren and Huffman, 1983). The complex refractive index $(n + im)$ of cloud droplets was configured as n=1.33 and m=1.56x10$^{-7}$ for liquid-water at 758 nm (Hale and Querry, 1973).

The line-by-line RT calculations in the $O_2$ $A$-band are computationally very demanding, and this precludes the deployment of on-line calls to VLIDORT during the processing of TROPOMI data. For this reason, RTM simulations for the range of S5P viewing conditions are performed in advance. Node points for the RTM are created using a "smart sampling" technique (Loyola et al., 2016) that minimizes the number of calls to the RTM and at the same time optimally covers the input space.

There are many millions of forward model calculations required; this process is done off-line and normally takes several weeks to complete. In the next step, the LBL simulations are convolved with the TROPOMI instrumental spectral response function and the resulting radiances are used to train a neural network that accurately approximates the RTM template output with a mean average relative error over the $O_2$ $A$-band spectral window for all scene geometries below one percent. The node point generation, RTM simulation, and neural-network training is done using the smart sampling and incremental function learning technique (Loyola et al., 2016). The input space (surface properties, cloud properties and geometries) is not sampled using a set of regular grids, but instead a smart sampling technique (Loyola et al., 2016) is used to optimize the distribution of multi-dimensional points within the (input) state space. The total number of computational nodes was of the order of some hundred thousand. The trained neural network that computes the $O_2$ $A$-band sun-normalized radiances is used in the UPAS operational environment, and this enables ROCINN retrievals to be done very quickly.

## 4.5. Inverse model

If $\mathbf{x}$ is the state vector $\{\tau_c, Z_{ct}, A_s, f_c\}$ comprising possible cloud parameters for retrieval, and $\mathbf{b}$ denotes a vector of auxiliary forward-model parameters (surface properties, viewing geometry, etc.), we write the measurement vector as $\mathbf{y}^\delta = \mathbf{F}(\mathbf{x}, \mathbf{b}) + \boldsymbol{\delta}$, where $\mathbf{F}$ is the forward model and $\boldsymbol{\delta}$ is the data error vector. The inverse problem defined by this equation is nonlinear and ill-posed, and regularization is required in order to obtain a solution with physical meaning. The degree to which the problem is ill-posed is partly characterized by the condition number $c(\mathbf{K}) = \gamma_{max}/\gamma_{min}$ of the Jacobian matrix $\mathbf{K} = d\mathbf{F}/d\mathbf{x}$, where $\gamma_{max}$ and $\gamma_{min}$ are the largest and the smallest singular values of $\mathbf{K}$, respectively.

In the form of Tikhonov regularization used here, the regularized solution $\mathbf{x}_\alpha^\delta$ minimizes the objective functional:

$$\mathfrak{F}_\alpha(\mathbf{x}, \mathbf{b}) = \tfrac{1}{2}\{\|\mathbf{F}(\mathbf{x}, \mathbf{b}) - \mathbf{y}\|^2 + \alpha\|\mathbf{L}(\mathbf{x} - \mathbf{x}_a)\|^2\}, \tag{9}$$

Here, $\alpha$ denotes the regularization parameter, and $\mathbf{L}$ is the regularization matrix (Doicu et al., 2010). The functional is defined with the $L_2$ Euclidean norm. The minimizer for Eqn. (9) can be computed with Gauss-Newton methods.

In statistical inversion theory, the Bayesian approach or the optimal estimation method can be regarded as a stochastic version of Tikhonov regularization. The maximum *a posteriori* solution coincides with the Tikhonov solution when the state vector $\mathbf{x}$ and the noise vector $\boldsymbol{\delta}$ are Gaussian random vectors with covariance matrices $\mathbf{C_x} = \sigma_x^2\mathbf{I}_n$ and $\mathbf{C_\delta} = \sigma^2\mathbf{I}_m$ respectively, where $\sigma_x$ and $\sigma$ are the corresponding standard deviations, and $\mathbf{I}_n$ is the identity matrix of size $n$. In this case, the regularization parameter $\alpha$ is the ratio of these two variances, that is, $\alpha = \sigma^2/\sigma_x^2$.

As noted above, the operational ROCINN algorithm with CAL (or CRB) retrieves two cloud parameters: the cloud top height and cloud optical thickness (or cloud albedo) with the *a priori* cloud fraction taken from OCRA and the surface albedo from the MERIS black-sky climatology at 760 nm (Popp et al., 2011). Note that the cloud fraction and the surface albedo are included in the state vector with a very strong regularization (i.e. only very small changes are allowed) in order to improve the fitting. These very small changes refer to the differences between the retrieved value of cloud fraction (and surface albedo) and their corresponding *a priori* value. The regularization parameter for cloud fraction and surface albedo is very high (i.e., two orders of magnitude higher than that for cloud top height and cloud optical thickness/cloud albedo) and thus, these retrieval parameters are always well within 1% difference from their *a priori* values. The state vector includes additionally a single wavelength registration shift parameter that takes care of the Doppler effect. The inverse model requires the partial derivatives of the radiances with respect to the state vector elements and these Jacobians are provided by the forward model.

Convergence is reached when either the residual $\|\mathbf{F}(\mathbf{x}, \mathbf{b}) - \mathbf{y}\|^2$ or incremental changes in the retrieved parameters $\Delta_{\mathbf{x}}$ are smaller than pre-defined values (defaults 5E-3 and 5E-5 respectively), or when the maximum number of iterations (default 50) is reached. The default value for the regularization parameter $\alpha$ is 1E-4.

## 4.6. Retrieval diagnostics

The equivalence between the Bayesian approach and the method of Tikhonov regularization enables us to analyze the information content of the signal with respect to the retrieved parameters in a stochastic framework (Schüssler et al., 2014). The degrees of freedom for signal (DFS) is a measure of the number of independent pieces of information in the measurement, and it gives the minimum number of parameters which can be used to define a state vector without loss of information. It is defined as the trace of the averaging kernel matrix, which represents the sensitivity of the retrieval to changes in the *true* state. The DFS can be computed as:

$$DFS = \Sigma_i^n \frac{\gamma_i^2}{\gamma_i^2 + \alpha},$$

(10)

where $\gamma_i^2$ are the singular values of the matrix $\mathbf{K}$.

Another useful criterion for the estimation of the retrieval quality is the Shannon information content (SIC), which is a measure of the incremental gain in information, defined as the entropy difference between the *a priori* and *a posteriori* states; the corresponding formula reads as:

$$SIC = \frac{1}{2} \Sigma_i^n \log\left(1 + \frac{\gamma_i^2}{\alpha}\right).$$

(11)

The accuracy of the regularized solution is represented by the mean square error matrix:

$$\boldsymbol{S_\alpha} = \varepsilon\left\{\left(\boldsymbol{x}^\dagger - \boldsymbol{x}_\alpha^\delta\right)\left(\boldsymbol{x}^\dagger - \boldsymbol{x}_\alpha^\delta\right)^T\right\} \approx (I_n - \boldsymbol{A}_\alpha)\left(\boldsymbol{x}_\alpha^\delta - \boldsymbol{x}_a\right)\left(\boldsymbol{x}_\alpha^\delta - \boldsymbol{x}_a\right)^T(I_n - \boldsymbol{A}_\alpha)^T + \sigma^2 \boldsymbol{K}_\alpha^\dagger \boldsymbol{K}_\alpha^{\dagger T},$$

(12)

where $x^\dagger$ is the exact solution or "true state", $x_\alpha^\delta$ the regularized solution, $x_a$ the *a priori* state vector, $A_\alpha$ the averaging kernel matrix, $K_\alpha^\dagger$ the generalized inverse, $\sigma$ the noise standard deviation, $\alpha$ the regularization parameter and $\varepsilon$ the expected value operator. Further information on the mean square error matrix and Tikhonov regularization can be found in (Doicu et al., 2010).

## 4.7. Retrievals using synthetic spectra

In order to evaluate the performance of the ROCINN retrieval algorithm in TROPOMI/S5P, a data set of synthetic TROPOMI measurements has been created. Synthetic spectra were computed using VLIDORT for a number of different scenarios characterised by various illumination and observation geometries, surface albedo and cloudiness. In particular, the following input space has been covered using the smart sampling technique: surface height [0 – 4] km, surface albedo [0 – 1], cloud top height [2 – 15] km, cloud optical thickness [2 – 50], viewing zenith angle [0 – 75] degrees, solar zenith angle [0 – 90] degrees, relative azimuth angle [0 – 180] degrees.

In general, these closed-loop ROCINN_CAL retrieval results are excellent; the cloud top-height results have no bias. The ROCINN_CRB retrieval was also applied to the same data set of synthetic spectra in order to obtain retrievals of the cloud height and cloud albedo. The results are shown in Figure 3; as expected, the CRB cloud height is systematically below the simulated cloud top-height, with a median difference of $1.2 \pm 0.4$ km.

Figure 4 shows the correlation from ROCINN_CAL retrievals of cloud optical thickness and ROCINN_CRB retrievals of cloud albedo. The differences in the cloud optical thickness are symmetrical about the 0-bias, whereas the differences in the cloud albedo are slightly skewed towards negative values.

## 5.  Error characterization

The accuracy of operational TROPOMI/S5P cloud products retrieved using OCRA and ROCINN is dependent on a number of different error sources.

The most important sources of model parameter uncertainty in ROCINN are errors on the assumed values for cloud fraction and surface albedo. Associated cloud property retrieval errors due to this source are discussed in detail in (Schüssler et al., 2014). Summarizing these findings, the cloud top height and cloud optical thickness can be accurately retrieved, even when the cloud fraction is underestimated or overestimated by as much as 20-30%. On the other hand, the cloud optical thickness retrievals are quite sensitive to uncertainty in the surface albedo. The sensitivity study from Schüssler et al., (2014) showed that deviations of ±10 % in the surface albedo introduce uncertainties of ±5 in the cloud optical thickness retrieval. The cloud retrievals are almost insensitive to cloud geometrical thickness uncertainties. In particular, for deviations of 50% in the cloud geometrical thickness, the retrieval errors in the cloud top height and cloud optical thickness are lower than 0.4 km and 2.0, respectively. Note that for the accurate retrieval of cloud geometrical thickness from the $O_2$ *A*-band either multi-angular (Merlin et al., 2016) or high spectral resolution (Richardson and Stephens, 2017) measurements are needed.

Errors due to forward-model uncertainty are the hardest to quantify, as these are due to sources such as mathematical discretization choices and physical simplifications. The most basic assumption is of course the use of a simplified 1-D radiative transfer model as mandated by the IPA. 3-D RTM of atmospheres with clouds is notoriously difficult and time-consuming. With the relatively small TROPOMI spatial footprint, horizontal inhomogeneity in cloud fields will be an important consideration from both the geometrical and the radiation perspectives. Some results for a 3-D treatment with clouds have been reported using Monte-Carlo models (Marshak and Davis, 2005) and more recently using stochastic RTM methods (Doicu et al., 2014). A detailed analysis of the uncertainties induced by the assumption of IPA with 1-D RTM can be found in a recently published paper (Efremenko et al., 2016); this analysis is the first of its kind to quantify 3-D forward model and retrieval errors in ozone and cloud properties derived from UVN measurements. The results from (Efremenko et al., 2016) indicate that the 1-D model generally underestimates radiances in the continuum of the oxygen *A*-band, while the radiances in the absorption peaks are basically the same. As a consequence, the cloud optical thickness is systematically slightly underestimated by retrievals based on the IPA, whereas the cloud-top height retrievals are generally unaffected. That paper also shows that use of the IPA leads to systematic errors in the retrieved ozone height-resolved partial columns.

The selection of a single liquid water cloud for the parametrization of clouds is a good approximation to describe light scattering by liquid cloud droplets in the atmosphere. The probability of a photon scattering in the forward direction is larger when a light beam at NIR wavelengths interacts with a cloud droplet with effective radius of a few µm, and thus the phase functions of water clouds can be well modelled by Mie theory (Kokhanovsky, 2004). However, Mie theory is not adequate to describe scattering by larger droplets of complex arbitrary shapes (e.g., ice crystals), and consequently the phase function of ice clouds cannot be modelled by spherical poly-dispersions. (Takano and Liou, 1989; Takano and Liou, 1995; Kokhanovsky, 2004). The associated errors can be assessed using radiative transfer simulations based on ice scattering phase functions as input to the ROCINN_CAL retrieval. The cloud optical thickness varies from 5 to 30 and the cloud height from 6 to 10 km. We have found that the retrieved cloud top height is almost unaffected, with an average error smaller than 2% for all cases considered. The errors on optical thickness are of the order of 1% for thick clouds and are larger for thin clouds with retrieved values of 6.06 compared to input optical thickness of 5.0.

Another source of error in the forward model is the simple assumption of a single cloud layer, since multi-layered clouds are often present in reality. The most common such situation in the atmosphere occurs with two cloud layers; one low-level cloud and a second mid-level cloud (Wang et al., 1999). With a view to obtaining estimates of the uncertainties of ROCINN to such double layer conditions, we have simulated the radiances for a double cloud layer model consisting of a mid-level cloud deck (Altocumulus/Altostratus) on top of the low-level cloud. The upper layer has a top height of 6 km, geometrical thickness  2 km and an optical thickness of 10 (Warren et al., 1985).

The first group of simulation tests is based on a situation with a thick low-cloud having cloud optical thickness of 25 and a cloud top height varying from 2 to 4 km below the mid-level cloud (see Figure 5). We found that the retrieved cloud optical

thickness is not affected by the position of the lower cloud, and the retrieved value was approximately 30. The accuracy of the cloud top height retrieval seems to depend on the separation distance between the two clouds (cloud bottom height of the upper cloud - cloud top height of the lower cloud). When the lower cloud is well separated from the upper cloud, the error in the retrieved cloud top height becomes larger (e.g., when the lower cloud has a top of 2 km, ROCINN_CAL retrieved a

cloud top height of approximately 4 km, and when the lower cloud has a top of 3.5 km, the retrieved value is 4.4 km). In the second group of simulations, the lower cloud has a top height of 2 km and optical thickness varying from 2 to 30 (see Figure 6). Now, the mean retrieved cloud top height from ROCINN_CAL was 4.3 km but with variations between 3.8 and 5.3 km. The large cloud top height values were retrieved for an optically thin lower cloud. However, in reality, with atmospheric scenes with two-layered clouds, it is common that the lower cloud is thicker than the upper one (Warren et al., 1985; Wang

et al., 1999). In such cases (lower cloud optical thickness > 20), the cloud top height retrieved with ROCINN_CAL is below 4 km. The retrieved cloud optical thickness is absolutely dependent on the cloud optical thickness of the lower cloud. For thick low-clouds (cloud optical thickness of 25-30), ROCINN retrieves a cloud optical thickness which basically corresponds to the cloud optical thickness of the lower cloud but with a small contribution of the second upper cloud (retrieved cloud optical thickness of 29-32).

In Figure 5, for all the simulations that were performed, the cloud height retrieved from CRB was about 0.8 km lower than that retrieved value from CAL. Moreover, as seen in Figure 6, the cloud height retrieved from ROCINN_CRB is even less sensitive to the two-layered cloud layer when the lower cloud is optically thinner. For very thin low-clouds (of optical thickness ~2), the difference between retrieved cloud heights from CAL and from CRB increased to as much as 1.6 km. Nevertheless, as stated above, when two cloud layers co-exist in the same atmospheric column, usually the low-level cloud is

optically thicker than the mid-level cloud (Warren et al., 1985; Wang et al., 1999).

From the TROPOMI calibration exercise, results have indicated that instrumental errors such as signal-to-noise and radiometric uncertainties in the UVN region are relatively small, although stray light issues were identified in the NIR band (NIR out-of-spectral band straylight analysis report, S5P-KNMI-OCAL-0152-RP, issue 0.1, 2017-05-11, in review). Both the errors induced on retrieved cloud properties due to NIR straylight and the precise knowledge of the slit function response

functions will be assessed when the instrument provides measurements from space. However, in the absence of real measurements, the impact of stray light in the Oxygen $A$-band has been initially assessed using a flat error of 1.2 % in the radiances (NIR out-of-spectral band straylight analysis report, S5P-KNMI-OCAL-0152-RP, issue 0.1, 2017-05-11, in review). The absolute errors on the cloud properties can be seen in Figure 7 for both CRB and CAL models. The retrievals from ROCINN_CRB are almost unaffected by the presence of stray light, with a mean error of 0.003 km for cloud height

and a mean error of 0.007 for cloud albedo. These errors induced in the cloud parameter retrievals by ROCINN_CAL are higher, with a mean cloud top height error of 0.3 km and a mean error of 1.0 for cloud optical thickness.

**5.1.  Co-registration inhomogeneity flag**

An important source of error is the spatial mis-registration between the UV/VIS and NIR bands from TROPOMI. The combination of information from different spectral bands is not as straightforward as it appears, since the spatial regions covered by the ground pixels from different spectral bands do not match exactly. One method for combining information from different bands is by means of the TROPOMI co-registration mapping tables, which contain the fractions of overlapping areas between the source and target pixels (Further information in the following document: Sentinel 5 precursor inter-band coregistration mapping tables, S5P-KNMI-L2-0129-TN, issue 4.0.0, 2015-11-23, released). For combinations based on OCRA UV bands 3, 4, 5 and ROCINN NIR band 6, a static co-registration table suffices. However, this method implies a smoothing of the source band products.

In this regard, a cloud co-registration inhomogeneity flag (CCIF) will be included in the S5P cloud products. This is determined as follows. First, a cloud co-registration inhomogeneity parameter (CCIP) is defined as the weighted averaged gradient of cloud fractions:

$$CCIP_j = \frac{\sum_i \omega_{ij} |f_{ci} - f_{cj}|}{\sum_i \omega_{ij}}, \tag{13}$$

where the weights $\omega_{ij}$ correspond to the co-registration mapping values between UV bands (source, index $i$) and the NIR band (target, index $j$). The CCIF is defined as:

$$CCIF_j = CCIP_j > p, \tag{14}$$

where $p$ is a fixed threshold which has been set to 0.4 as the baseline, following extensive testing using the Suomi-NPP VIIRS cloud product resampled to the TROPOMI spatial grid.

## 6. Application to OMI and GOME-2 and comparison with independent retrievals

The operational OCRA and ROCINN cloud algorithms presented in this paper have been fully implemented and tested in the TROPOMI/S5P operational processor UPAS under development at DLR. The resulting output files will follow the same netCDF format structure used for all the S5P L2 products. The main outputs are the cloud products retrieved with OCRA and ROCINN_CAL, while the ROCINN_CRB retrievals are to be reported in the detailed results group. For more information, the reader is referred to the S5P Cloud Product User Manual (Pedergnana et al., 2016).

In this section we present the results obtained through application of the TROPOMI/S5P cloud algorithms implemented in UPAS to measurements from OMI and GOME-2.

### 6.1. Comparison of OCRA_RGB and OCRA_GB using GOME-2

The two-colour OCRA_GB model was tested against the three-colour OCRA_RGB model. The overall median cloud fraction difference is 0.026 and the mean 0.032 for a single day of GOME-2A measurements from July 1[st], 2012. For cloud fractions smaller than 0.1 the median and mean differences are 0.007 and 0.015 respectively.

Figure 8 shows the cloud fraction differences in a box-and-whisker plot. The bottom, inside-band, and top of the box are the first, median, and third quartiles of the differences.

## 6.2. Comparison of OCRA with OMI and MODIS cloud fractions

As part of the S5P project we have adapted the OCRA algorithm to the OMI sensor (the precursor of TROPOMI).

OMI is a nadir-viewing push-broom spectrometer observing solar backscatter radiation in the ultraviolet and visible wavelength range up to 500 nm (Levelt et al., 2006). The swath width is 2600 km on ground, encompassing more than 60 across-track pixels. The highest spatial resolution of 13 x 24 km$^2$ (in normal mode) is achieved for the nadir pixels. OMI was launched in 2004 on the NASA Aura satellite platform.

The first step is the calculation of a monthly OCRA cloud-free background data set as described in section 3.2; this was based on 3.5 years of OMI measurements from January 2005 to June 2008. The OMI cloud-free background data set for the month of August is shown in Figure 9. Note that OMI is extremely stable, with almost no instrument degradation - this facilitates significantly the calculation of the OCRA cloud-free backgrounds.

The temporal resolution is one month, i.e. for a given grid cell, all measurements from a given month are aggregated in order to reflect seasonal surface variations. Finally, all data from the years 2005, 2006, 2007 and 2008 are considered for the final monthly maps. The spatial grid resolution is 0.2 degrees in latitude and 0.4 degrees in longitude. The cloud-free reflectance value for any given time and geolocation is found via linear interpolation between the two adjacent monthly cloud-free maps. This linear interpolation between monthly maps was found to give the best trade-off between the necessity to have as many measurements as possible per grid cell in order to ensure a cloud-free situation (i.e. a long timescale is desired), and the requirement to be sensitive to rapid changes in the surface conditions such as snowfall or melting (i.e. a short timescale is desired).

Secondly, the scaling and offset factors are computed following the procedure described in section 3.3. The resulting scaling factors are $\alpha_B = 2.88$ and $\alpha_G = 2.14$, and the offset factors are $\beta_B = 0.0138$ and $\beta_G = 0.0180$.

For the comparison with MODIS, we used the following cloud products:

- The OMAERUV product, provided by O. Torres. The cloud fraction is an ancillary product from the absorbing aerosol index algorithm based on OMI radiances at 388 nm and 354 nm (Torres et al., 2007).

- The OMCLDO2 product version 2.0, provided by P. Veefkind. The cloud fraction of this product is based on the OMI $O_2$-$O_2$ absorption feature around 477 nm (Veefkind et al., 2016).

- The OMCLDRR product, taken from the OMMYDCLD product version 003 (J. Joiner 2014, OMI/Aura and MODIS/Aqua Merged Cloud Product 1-Orbit L2 Swath 13x24 km$^2$ V003, Greenbelt, MD, USA, Goddard Earth Sciences Data and Information Services Center (GES DISC), Accessed 26 Oct 2016). This OMI cloud fraction is derived at 354 nm where the contribution of Raman scattering is minimal.

- The MODIS product co-located to OMI footprints from the OMMYDCLD product version 003. This product provides the OMI/Aura and MODIS/Aqua merged cloud products. No MODIS/Terra data are incorporated here. Since both Aura and Aqua are part of the A-train, the overpass times are comparable (the separation between Aura and Aqua is 8 minutes).

Figure 10 shows the global cloud maps obtained with OCRA, OMAERUV, OMCLDO2, and OMCLDRR from OMI measurements on July 16[th], 2005; the four algorithms generate similar cloud features.

A quantitative comparison of the zonal mean cloud fractions from OMI and those derived with MODIS is presented in Figure 11. The UV sensors are less sensitive to optically thin clouds than thermal infrared sensors and as expected, MODIS generates larger cloud fractions compared to those from OMI. The cloud fractions of OMCLDO2 and OMCLDRR are

similar because both algorithms assume a fixed cloud albedo or reflectance of 0.8 for the retrieval, but overall the cloud albedo is significantly smaller (e.g. Lelli et al., 2012 report a mean global cloud albedo value of 0.63 based on GOME data from 1996-2003) and therefore the retrieved effective cloud fractions are significantly smaller than the MODIS geometrical cloud fractions. On the other hand, OCRA and OMAERUV do not need to assume a fixed cloud albedo, and their retrieved cloud fractions are larger than those from OMCLDO2 and OMCLDRR. OCRA and OMAERUV report cloud fraction values

more representative of the radiometric cloud fraction based on TOA radiances measured by the instrument. We should emphasize here that a direct comparison between the MODIS geometric cloud fraction and the OMI-derived radiometric or effective cloud fractions should be treated with caution.

### 6.3. Comparison of ROCINN cloud top height and optical thickness from GOME-2

In preparation for S5P we have applied the ROCINN_CAL algorithm to GOME-2, and in this section we present for the first time the resulting cloud parameter retrievals.

The Global Ozone Monitoring Experiment-2 (GOME-2) is a nadir-viewing optical spectrometer that senses Earth's backscattered radiance and solar irradiance at UV/VIS/NIR wavelengths in the range 240-790 nm (Munro et al., 2016). The nominal full GOME-2 swath has a width of 1920 km in the direction perpendicular to the flight direction and a single scan

line has an extension of 40 km in the flight direction. The ground pixels have a spatial resolution of 80 x 40 $km^2$. In addition, broad-band Polarization Measurement Devices (PMDs) provide an eight-fold higher spatial resolution, i.e. 10 x 40 $km^2$ for a selection of 15 spectral windows. Currently there are two GOME-2 operational sensors onboard the EUMETSAT MetOp-A and MetOp-B satellites launched in 2006 and 2012 respectively; both GOME-2 sensors are operated in tandem providing global measurements on a daily basis. A third GOME-2 sensor onboard MetOp-C will be launched in 2018.

The VLIDORT line-by-line RTM simulations described in section 4.4 are convolved with the GOME-2 instrumental spectral response functions and the results used to train a neural network that accurately approximates the $O_2$ $A$-band reflectances

(Loyola et al., 2016). As noted in section 4.5, the ROCINN cloud top-height and optical thickness are retrieved using the Tikhonov inversion, taking as input the OCRA cloud fraction computed from the GOME-2 PMD measurements (Lutz et al., 2016).

Figure 11 shows the global cloud maps obtained with ROCINN_CAL and ROCINN_CRB from GOME-2A (GOME-2 on MetOp-A) measurements taken on July 1$^{st}$, 2012. As expected from the retrievals with synthetic data (section 4.7) the cloud height retrieved using ROCINN_CRB is smaller than that retrieved using ROCINN_CAL; the CRB model retrieves the centroid of the cloud and not the cloud-top (Joiner et al., 2012)). The cloud optical thickness from ROCINN_CAL nicely correlates with the cloud albedo from ROCINN_CRB.

The histogram of absolute differences between the GOME-2A cloud heights on July 1$^{st}$, 2012 obtained with ROCINN_CAL and ROCINN_CRB is presented in Figure 13. The CRB model underestimates the cloud top height with a median difference of $0.92 \pm 0.75$ km. +/- These results are consistent with the retrievals obtained using synthetic data from section 4.7.

The diagnostic quantities DFS and SIC for the test case of July 1$^{st}$, 2012 were found to lie in the ranges 1.2-4 and 2-11 respectively for ROCINN_CAL. These values are lower than those obtained for ROCINN_CRB (DFS between 2.1 and 4.3, SIC in the range 4-17), showing that the CAL-retrieved cloud quantities depend more on the *a priori* information.

## 7. Conclusions

We have presented the latest versions of the retrieval algorithms OCRA and ROCINN to be used for the generation of the operational TROPOMI/S5P cloud products: cloud fraction, cloud top height (pressure) and optical thickness (albedo).

In UPAS, a special effort has been directed to optimizing the run-time performance of the algorithms in order to cope with the "big data" expected from TROPOMI (around 21 million ground pixels daily, with 1.5 million pixels per orbit). The operational cloud retrievals are extremely fast and accurate: the OCRA cloud fraction is computed using a simple expression (Eqn. 3), while the time-consuming generation of the cloud-free composite is done off-line. Similarly the complex and computationally expensive line-by-line RTM calculations needed for the ROCINN retrieval of cloud top height and optical thickness are replaced by fast artificial neural networks trained using the smart sampling and incremental function learning techniques (Loyola et al., 2016).

The OCRA and ROCINN algorithms are integrated in the S5P operational processor UPAS for the generation of near-real-time and off-line products. In this paper we have shown that UPAS cloud properties retrieved from OMI and GOME-2 measurements provide a good basis for anticipated retrievals from TROPOMI measurements themselves.

The algorithms presented in this paper will be used during the S5P commissioning phase. The operational TROPOMI cloud products will be validated using ground-based measurements of cloud radar and microwave radiometer instruments available

in CloudNet stations, and using cloud products from VIIRS (Visible/Infrared Imager and Radiometer Suite) onboard the Suomi NPP (NPOESS Preparatory Project) satellite of NASA/NOAA; the S5P orbit will trail five minutes behind Suomi NPP.

A number of future algorithm developments are planned once the TROPOMI data becomes available after the S5P launch:
the spatial mis-registration between the UV/VIS and NIR bands will be characterized and the possibility of a correction will be investigated, the effects of TROPOMI straylight in NIR on the cloud retrievals will be analysed, and a cloud-free background data set based on TROPOMI/S5P will be generated once the first full year of measurements becomes available; this will replace the initial cloud-free background data based on OMI.

The OCRA and ROCINN cloud parameters will be used for enhancing the accuracy of the operational TROPOMI/S5P trace
gas products total ozone (Loyola et al., 2017), formaldehyde, and sulphur dioxide (Theys et al., 2016). OCRA and ROCINN will be also used for the generation of operational cloud products from the geostationary Copernicus atmospheric composition mission Sentinel-4 (S4). In this way, cloud products from atmospheric composition missions S5P and S4 will be consistent, and together they will extend for the next two decades the unique UVN cloud data record (Loyola et al., 2010) initiated over twenty years ago with GOME/ERS-2.

**Acknowledgements**

Two co-authors are no longer affiliated to DLR: Sebastián Gimeno García (now at EUMETSAT, Darmstadt, Germany) and Olena Schüssler.

Thanks to the S5P L2WG cloud verification team, in particular Holger Sihler (MPIC) and Luca Lelli (IUP-UB) for helpful
comments. Thanks to Omar Torres (NASA), Joanna Joiner (NASA), and Pepijn Veefkind (KNMI) both for providing the OMI and MODIS cloud data used in our OCRA comparisons, and also for a number of helpful discussions. Thanks to KNMI/NASA for the OMI level 1 products and EUMETSAT for the GOME-2 level 1 products used in this paper.

This work has been performed in the framework of the TROPOMI/S5P project. We acknowledge financial support from Bayerisches Staatsministerium für Wirtschaft und Medien, Energie und Technologie (Grant 0703/89373/15/2013) and from
DLR programmatic (S5P KTR 2 472 046) for the S5P algorithm development, and from the ESA S5P L2WG project (Contract No.: 4000107711/13/NL/IB) for the S5P operational processor development.

Finally, the authors thank three anonymous reviewers for their fruitful inputs and suggestions which have helped to improve this manuscript.

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

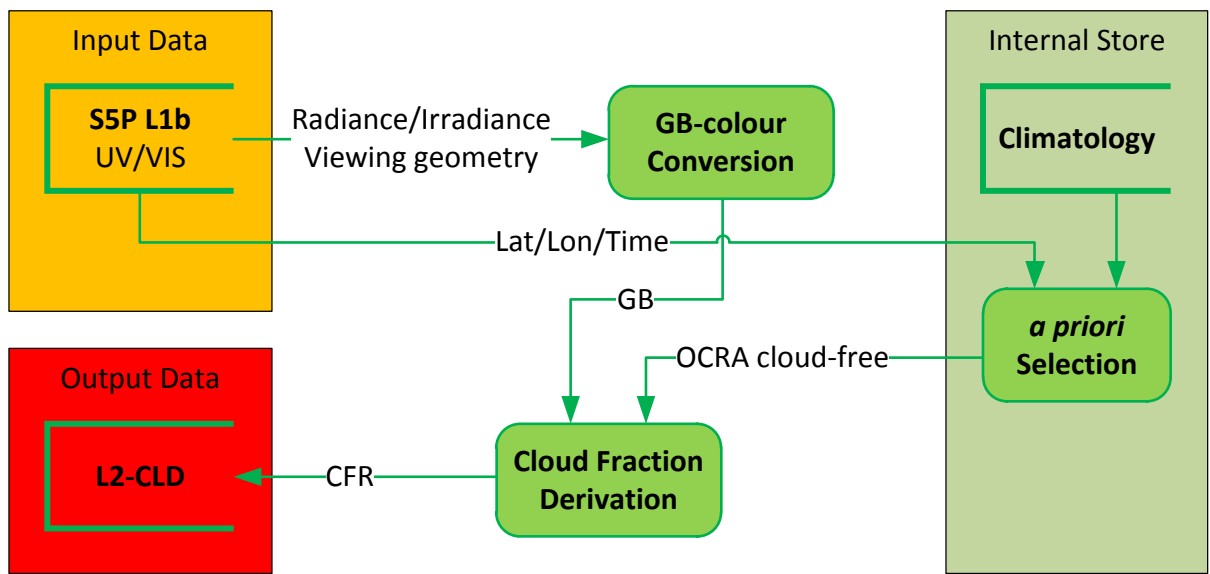

**Figure 1: Flow diagram for the OCRA algorithm for the retrieval of the radiometric cloud fraction (CFR).**

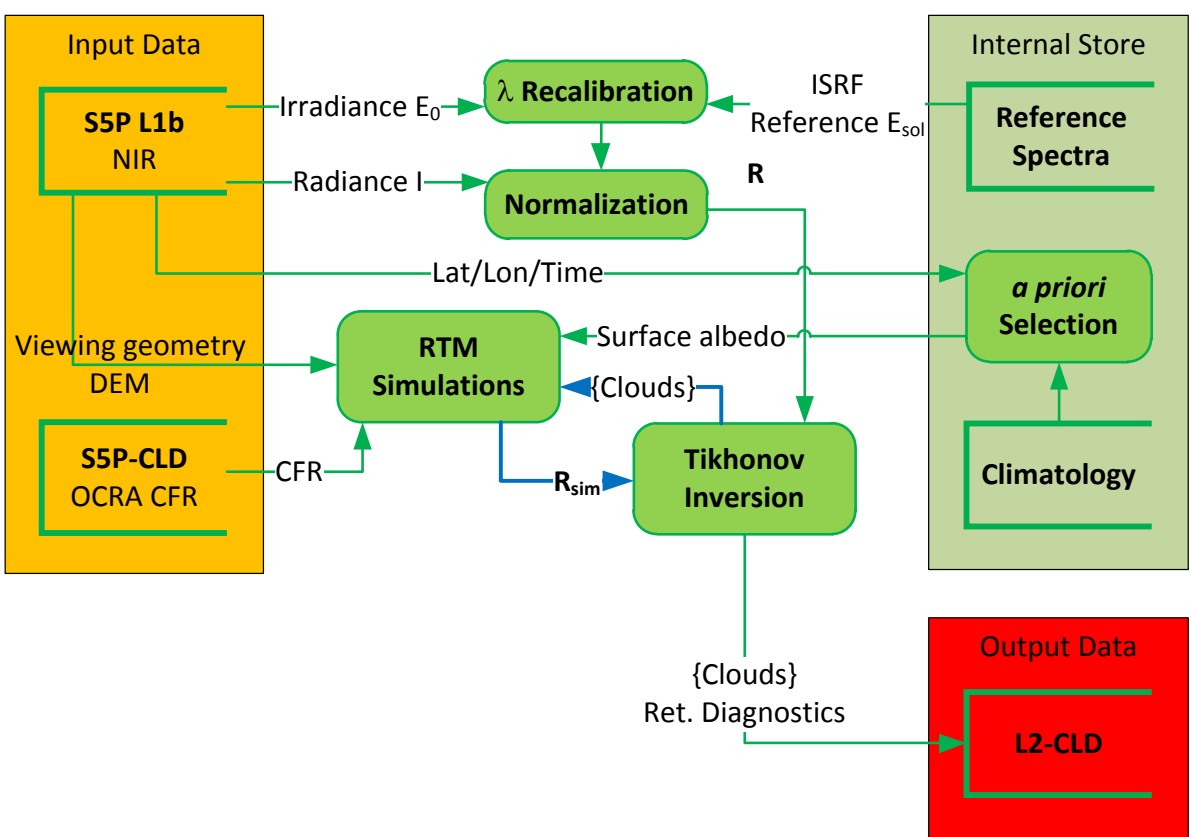

**Figure 2: Flow diagram for the ROCINN algorithm for retrieval of cloud properties. The blue arrows mark an iterative loop.**

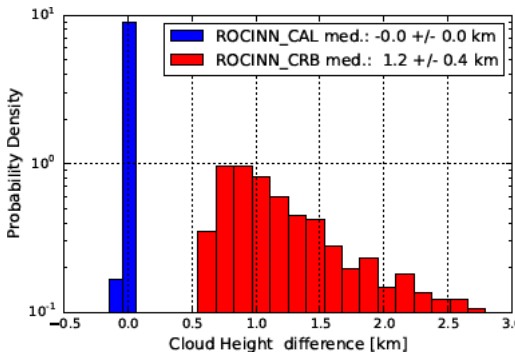

**Figure 3: Histogram of the absolute differences between the simulated spectra and the cloud height retrievals from ROCINN_CAL and ROCINN_CRB.**

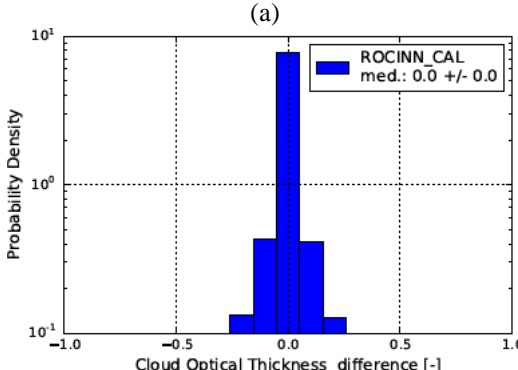 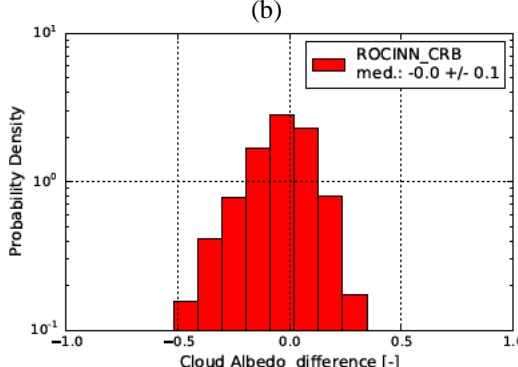

**Figure 4: Histogram of the absolute differences between the simulated spectra and (a) the cloud optical thickness retrievals from ROCINN_CAL and (b) the cloud albedo retrievals from ROCINN_CRB. The retrieved cloud optical thickness varies from 2 to 50, with the cloud albedo ranging from 0 to 1.**

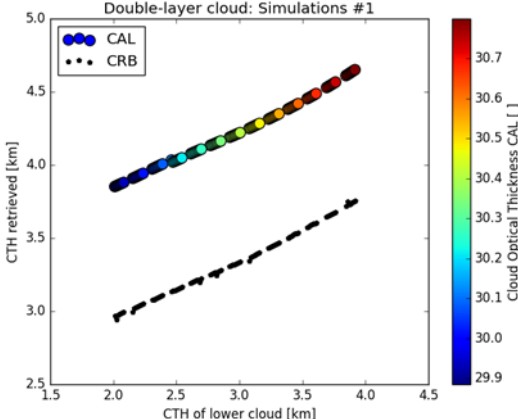

**Figure 5: ROCINN retrieved cloud properties for the first group of simulations (i.e., a low cloud with cloud optical thickness of 25, cloud geometrical thickness of 1 km and a cloud top height in the range 2-4 km, plus a mid-level cloud having cloud optical thickness of 10, cloud top height 6 km and cloud geometrical thickness 2 km).**

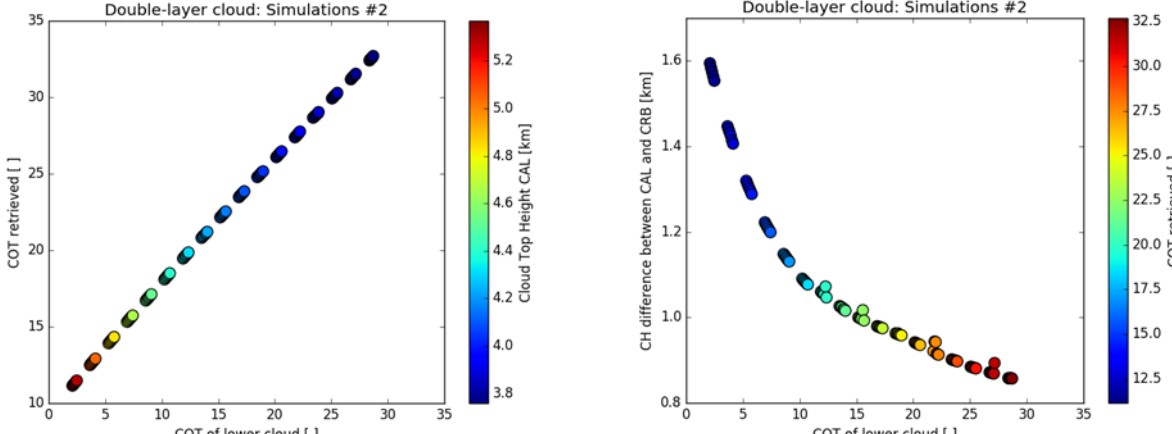

**Figure 6: (a) Retrieved cloud top height and cloud optical thickness from ROCINN_CAL for the second group of simulations (i.e., a low cloud with cloud top height of 2 km, cloud geometrical thickness 1 km and a cloud optical thickness in the range 2-30, plus a mid-level cloud with cloud optical thickness of 10, cloud top height 6 km and cloud geometrical thickness 2 km). (b) Differences in cloud height retrieval between CRB and CAL as a function of the optical thickness of the low cloud.**

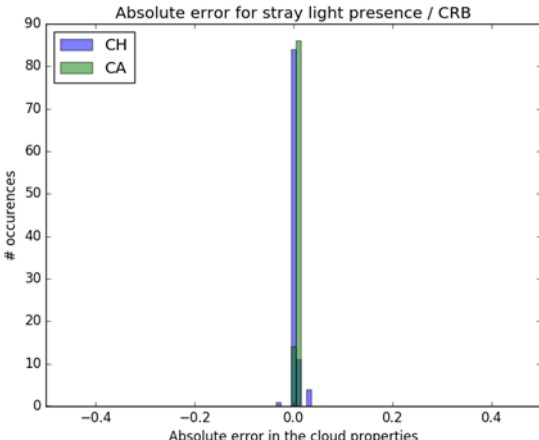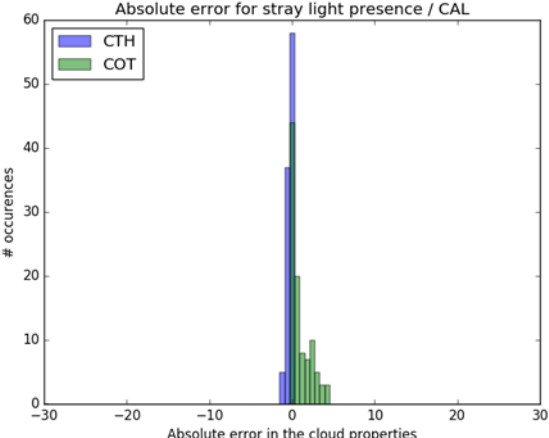

**Figure 7: Absolute errors in the cloud properties due to the presence of stray light: (a) cloud height and cloud albedo errors for CRB and (b) cloud top height and cloud optical thickness errors for CAL.**

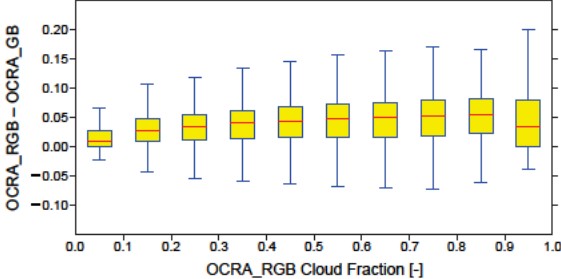

**Figure 8: Box-and-whisker plot of the OCRA cloud fraction difference using RGB and GB colours retrieved using GOME-2 data from July 1$^{st}$ 2012. The yellow boxes show the inter quartile range and median values are indicated as red lines. The overall median difference is 0.025.**

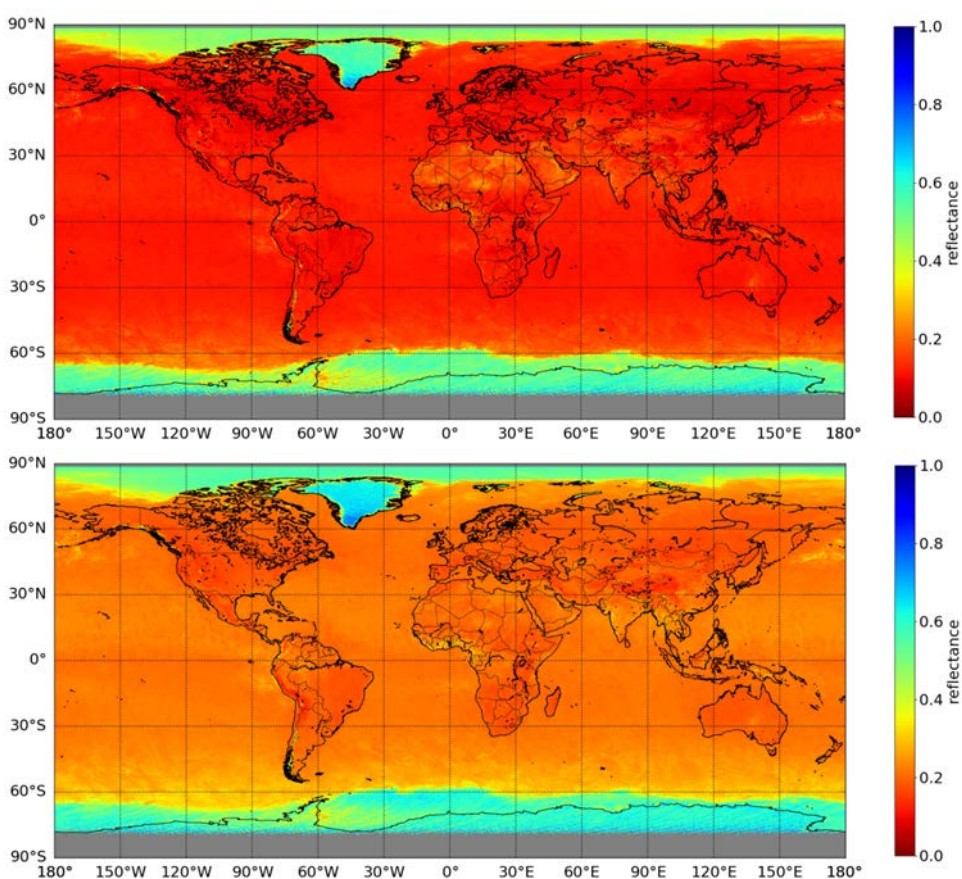

**Figure 9: OCRA cloud-free background for G (top) and B (bottom) reflectances calculated using OMI data from August.**

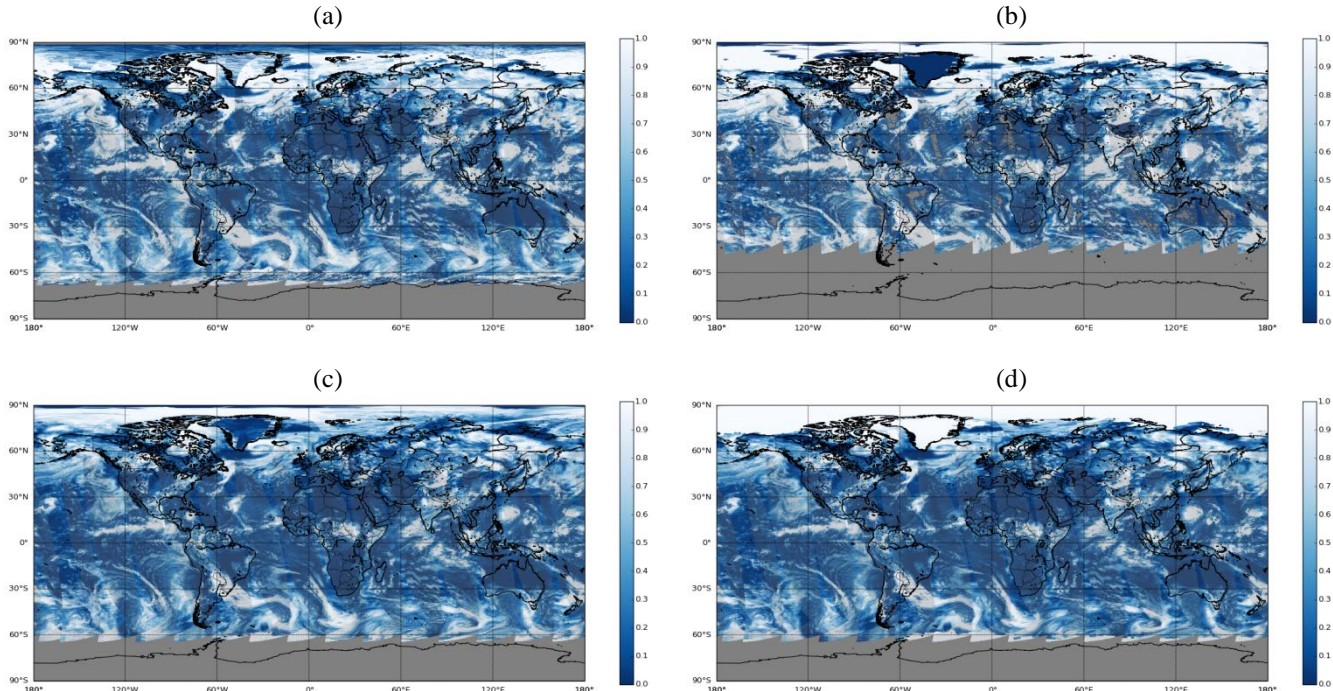

**Figure 10: Cloud fraction retrieved with (a) OCRA, (b) OMAERUV, (c) OMCLDO2, and (d) OMCLDRR from OMI measurements on July 16th, 2005.**

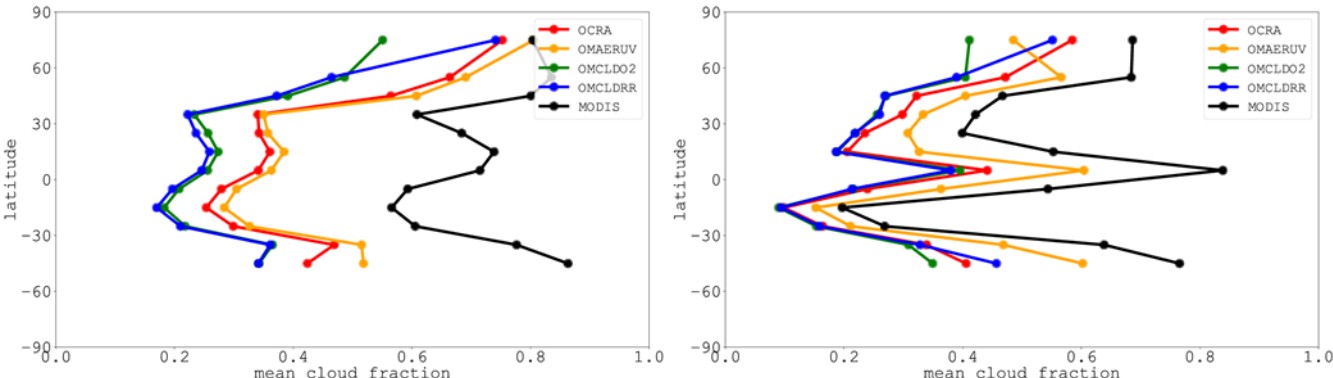

5    **Figure 11: Comparison of cloud fraction zonal means for results from the four OMI algorithms as seen in Figure 10 and from the MODIS measurements regridded to the corresponding OMI ground pixels. The left panel shows only measurements over ocean, while the right panel shows only measurements over land.**

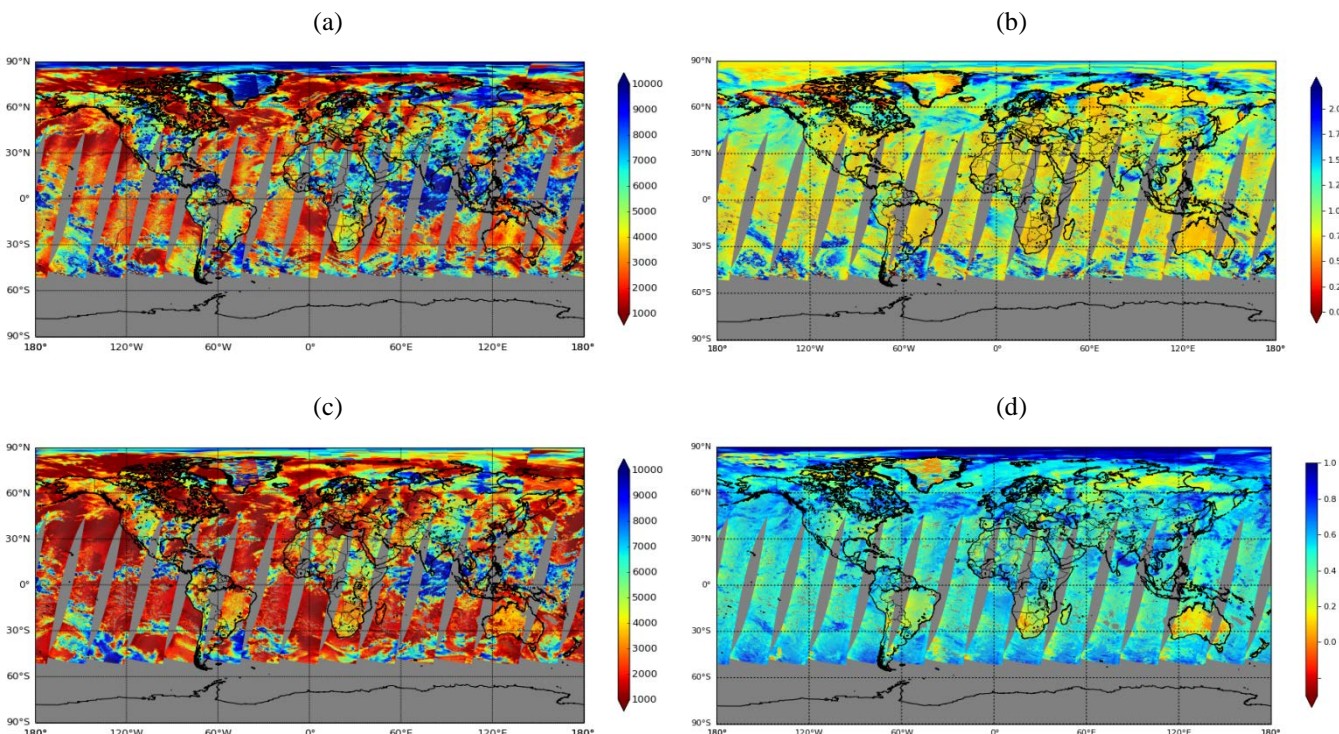

**Figure 12: Cloud top height (a) and cloud optical thickness (b) retrieved with ROCINN_CAL, cloud height (c) and cloud albedo (d) retrieved with ROCINN_CRB from GOME-2A measurements on July 1$^{st}$, 2012. The cloud height is displayed in meters and a logarithmic scale is used for the optical thickness.**

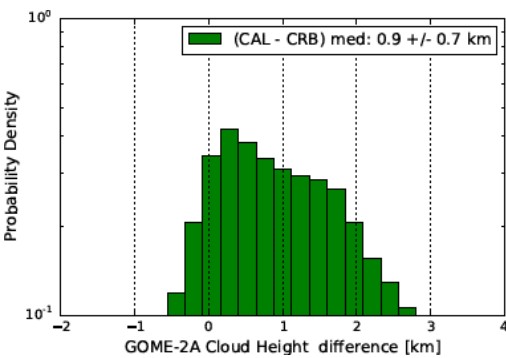

**Figure 13: Histogram of the absolute differences between the GOME-2A cloud heights derived from ROCINN_CAL and ROCINN_CRB as seen in Figure 12.**

