# Peer review of "The operational cloud retrieval algorithms from TROPOMI on board Sentinel-5 Precursor"

_Atmospheric Measurement Techniques, 2017_

## Referee Comment (RC1) · Anonymous Referee #2 · 30 Jun 2017

The manuscript describes the OCRA cloud fraction and ROCINN cloud pressure algorithms and their modifications that were made to adapt the algorithms for TROPOMI. Most material of the manuscript has been published. However, the paper contains some original material that is mainly related to the modification of the ROCINN algorithm. This material may be of interest for the developers of cloud algorithms for satellite hyperspectral radiometers. Moreover, the OCRA and ROCINN cloud algorithms were selected for TROPOMI; that is why it is important to document the algorithms in the literature. The paper subject is appropriate to AMT. Earlier work is recognized and credited. The abstract provides a sufficiently complete summary of the paper. The paper is well organized. I think that the paper needs significant revisions before recommending it for publication. The authors should address the following comments.

General comments

1. The authors state that the cloud pressure algorithm, ROCINN_CAL, provides better cloud-top retrievals than ROCINN_CRB. The TROPOMI cloud products are intended to use in trace-gas retrievals. It is not obvious that the cloud-top pressures can produce better trace-gas retrievals. The Mixed Lambertian Equivalent Reflectivity (MLER) model compensates for photon transport within a cloud by placing the Lambertian surface somewhere in the middle of the cloud instead of at the top. As clouds are vertically inhomogeneous, the pressure of this surface does not necessarily correspond to the geometrical center of the cloud, but rather to the so-called optical centroid pressure (OCP). Cloud OCPs are the appropriate quantity for use in trace-gas retrievals from satellite instruments. Cloud-top pressures are not equivalent to OCPs and do not provide good estimates of solar photon path lengths through clouds that are needed for trace-gas retrievals from ultraviolet and visible wavelength solar backscatter measurements (Ziemke et al., 2009; Joiner et al., 2012). The authors should prove that the ROCINN_CAL cloud-top pressures do produce better trace-gas (e.g. $O_3$ or $NO_2$) retrievals. That is particularly important in the view that the TROPOMI $NO_2$ algorithm makes use of OCPs from the MLER-based FRESCO+ cloud algorithm (van Geffen et al., TROPOMI ATBD of the total and tropospheric $NO_2$ data products, URL: https://sentinel.esa.int/web/sentinel/user-guides/sentinel-5p-tropomi/document-library, 2016).

J.R. Ziemke, J. Joiner, S. Chandra, P.K. Bhartia, A. Vasilkov, D.P. Haffner, K. Yang, M.R. Schoeberl, L. Froidevaux, and P.F. Levelt, Ozone mixing ratios inside tropical deep convective clouds from OMI satellite measurements, Atmos. Chem. Phys., 9, 573-583, 2009.

2. The OCRA algorithm has been described in detail in Loyola et al. (2007) and Lutz et al. (2016). In those papers, the authors used the normalized RGB (red-green-blue)

representation of colors. In this manuscript the authors propose the Green-Blue color system for TROPOMI. This switching to the GB system should be explained because the red channel (675-775 nm) is available in TROPOMI. The authors should also compare cloud fraction retrievals from RGB and GB using e.g. GOME-2 data.

3. The use of the Mie scattering model of clouds is new in the ROCINN algorithm. That is why it is important to show that the selection of a single water cloud model, i.e. a single phase scattering function, is representative. Clouds can be multilayer and vertically-extended. This significantly affects photon path lengths and thus oxygen absorption in the cloud. The authors should show that their selection of a vertically-uniform cloud model with a single geometrical thickness of 1 km is sufficiently representative. The authors should provide an estimate of possible cloud pressure errors associated with the selection of the cloud model.

4. The authors should add a couple of paragraphs describing how their radiometric cloud fraction is used in the DOAS trace-gas algorithms. It is important to highlight the differences between the use of the radiometric cloud fraction and effective cloud fraction that comes from the MLER model.

5. The authors show a comparison of the OCRA radiometric cloud fraction with the MODIS geometrical cloud fraction. It is unclear why the authors do not carry out a similar comparison of the ROCINN_CAL cloud-top pressure with the MODIS cloud-top pressure. This comparison should be done and quantitative results of the comparison should be provided.

Specific comments:

Abstract. Some numbers characterizing the error budgets are strongly recommended in the abstract.

Introduction. Please add the following reference and discuss how your approach differs from that by Diedenhoven et al. (2007)

Diedenhoven et al., Retrieval of cloud parameters from satellite-based reflectance measurements in the ultraviolet and the oxygen A-band, JGR, 112, D15208, doi:10.1029/2006JD008155, 2007.

P.4, L.13. "the minimum Lambertian equivalent" should be "the mixed Lambertian equivalent"

P.4, L.17. "in the range 330-390 nm" is incorrect; OMAERUV makes use of just two wavelengths 354 and 388 nm.

P.5, L.27. Is it correct that the scaling and offset factors are determined using daily satellite measurements, not monthly?

P.6, L.5. "a simplified sun-glint correction". Do you mean "sun-glint flagging"? Please provide information about the performance of the cloud algorithms over the sun glint area. For instance, this information can include the cross-track dependence of daily-averaged OMI cloud fraction and cloud pressure for such areas.

P.8, L.21-22. Your statement about small effect of the cloud phase (water or ice) should be proven by radiative transfer simulations. Please provide comparisons of computed TOA radiances for water and ice clouds and corresponding cloud pressure errors.

Section 4.4. Please provide information about a number of computational nodes over surface reflectance, surface altitude, solar and viewing angles.

P.9, L.11-12. Please provide typical errors of replacing exact radiative transfer simulations by neural network calculations for different sun-view geometries.

P.10, L.2-3. "the surface albedo climatology"; please provide a reference

P.10, L.4. "only very small changes are allowed" is a qualitative statement. Please provide quantitative information.

Section 4.6. describes well known theoretical estimates of the DFS and retrieval errors. Why the numerical estimates are not used in the text? I would remove this section and

retain just a reference.

Section 5. Most statements in this section are qualitative like "can be accurately retrieved" (L.18), "quite sensitive to", "less significant are ROCINN errors" (L.20). The section titled "Error characterization" should provide quantitative information.

P.11, L.19. Do you really mean "cloud geometrical fraction", not radiometric?

P.12, L.4. Why the NIR stray light effects "will be assessed when the instrument provides measurements from space"? You say that "stray light issues were identified in the NIR band". The stray light contribution can be important for most absorption lines of the oxygen A-band. The authors should assess the stray light effects on the retrieved cloud properties. It seems to be straightforward to simulate stray light and investigate the impact on cloud pressure retrievals.

P.13, L.21-22. "This OMI cloud fraction is based on the filling-in of solar Fraunhofer lines caused by Raman scattering" is incorrect. The OMCLDRR cloud fraction is derived at 354 nm where the Raman scattering contribution is minimal.

P.13, L.27. Please explain why "the UV sensors are not sensitive to optically thick clouds". What physics do you mean in this statement?

P.14, L.2-3. Please explain the meaning of "OCRA and OMAERUV report cloud fraction values more representative of the radiometric cloud fraction measured by the instrument". The instrument measures TOA radiances.

---

## Referee Comment (RC2) · Anonymous Referee #1 · 30 Jun 2017

This paper accomplishes two tasks: first, it provides a scientific update of a cloud retrieval algorithm and, second, it summarizes the suite of cloud products that will be available upon the launch of the Sentinel-5 platform with the TROPOMI payload.

As such, the readership can be wide and mainly made of two groups of individuals: experts in the field of cloud remote sensing and users of future TROPOMI data. This not only sets higher-than-usual requirements on the amount and quality of information to be conveyed in such a paper, but also demands a mixture of technical and scientific writing style.

In fact, while many concepts can be understood by the expert, users might not have the expertise and the required knowledge to understand the paper, especially when it goes down to error budgets and physical reasoning that support the conclusions of the

presented work.

Based on the above reasons, I think that this potentially important paper can be greatly improved with respect to readability and scientific information and publication should be warranted upon **major revisions**.

**Specific comments**

**Abstract**
The sentence provided on the higher accuracy of cloud properties derived from the NIR as compared to the TIR is indeed correct, but it is misleading for the reader, because she/he can think that a NIR-TIR comparison is one of the topic of the paper, which is not. So, this statement suits best as part of the introduction or the outlook, but not in the abstract, where, in my opinion, only an objective summary of the main matter of the paper should be given.

As an interesting topic on its own (the NIR-TIR comparison), I flag to the authors that within Cloud_cci (Stengel et al., 2017), TIR retrievals from (A)ATSR are compared with retrievals derived from a combination of TIR and the NIR oxygen A-band channels by MERIS (Fig. 8, p. 21, third panel from above, CTP [hpa]). As outcome, one can appreciate that the addition of the oxygen A-band from MERIS corrects for photon penetration depth issues of the TIR channels and the found average bias amounts to approx 60 hpa, which translates to approx 0.8 km.

Consistently, one recent study (Lelli et al. 2016) compared cloud properties derived from the oxygen A-band with the TIR-derived cloud heights of AATSR. It can be seen that TIR cloud retrievals are indeed placed lower (as the ROCINN_CRB) than the ones derived from the NIR with a scattering cloud layer model, as the ROCINN_CAL, by an average amount, again, in range 0.6 - 1.0 km.

Once the accuracy of ROCINN_CAL and ROCINN_CRB will be assessed, it becomes reasonable to state that the oxygen A-band delivers more accurate cloud heights than

the ones from TIR channels (albeit uncorrected).

**Section 1, Introduction, p 2, l 8-12**
Keeping in mind, as research outlook, the impact that a change of the used cloud model in cloud retrieval algorithms can have on the accuracy of retrieved trace gas columns, I appreciate a more detailed presentation of past work (facts and figures) in the field. In fact, the sentence "These studies have shown that cloud fraction . . . " is too general knowledge and does not properly convey the importance of the issue to be tackled.

It could be also somehow inaccurate, because when looking at du Piesanie et al (2013), the authors assessed the accuracy of SCIAMACHY water vapor columns as function of changing cloud fraction, optical thickness and cloud top height. They found that, using a scattering cloud model and the OCRA cloud fraction (making their results even more appropriate for this paper), CTH is the most critical parameter for water vapor, while cloud fraction and optical thickness are somewhat less relevant.

So, please, expand this paragraph, briefly reporting past results about trace gas accuracy and information on the cloud model assumption that has been respectively used to derive them (wherever available and appropriate).

**Introduction, p2, l13**
Is the spatial resolution the same for all TROPOMI bands? If not, please, report the correct information and briefly discuss how different footprint sizes can influence a joint exploitation (e.g., UV-Vis-NIR and SWIR).

**Introduction, p2, l17**
Overpass time of the mentioned sensors? This is important for the extension of the data record, as different sensing times will record different atmospheres.

**Section 2, p3, l5**
What is OCRA CF needed for as "baseline input"?

**p3, l10**

It is said the the ROCINN_CAL is here presented for the first time. Then one might wonder where was the ROCINN_CRB model presented? Please, provide reference.

**p3, l10-17**
This paragraph needs additional details on the errors as function of CRB/CAL, on the same line of thoughts of the impact of the cloud model on the accuracy of trace gases.

**Section 2.1, p3, l24**
References for ROCINN algorithm?

**p3, l27-28**
Two aspects are not clear here. (1) why the IPA allows 1-D plane parallel RT of cloud-contaminated scenes and (2) whether the previous statement also holds for future TROPOMI measurements due to 3-D effects. Please, discuss this aspect.

**p4, l6**
PMD-derived cloud fraction benefits not only of the spectral coverage but also of a spatial resolution finer than the science channels. So, please, mention this.

**p4, l10**
The heritage OMI cloud fraction algorithms need a bit more details to make the reader understand how the cloud detection works. I might understand it, but it is not something all readers can follow.

**Section 3, p4, l22**
Figure 1 contains a block which is not properly described in none of the following sub-sections, the "internal store". The authors need to address (and amend the manuscript where appropriate) the following questions: (1) Why the need of a-priori selection if the brightness criterion should already deliver a minimum reflectance?
(2) What is the climatology used for?
(3) What climatology? Source, time-space aggregation? Quality of the values? Is a climatology appropriate and does it have shortcomings for the task?

**Section 3.1, p4, l28**
It's the first time I read the terminology "ground-cover projection". What is this?

**Section 3.2, p5, l8**
It is said that reflectances are independent of atmosphere and line-of-sight. What do aerosol, Rayleigh and the surface do? Especially for the latter, does surface reflectivity change over the time needed to build the composite? This is crucial, especially when thinking at a small footprint. Please, add information on the impact of these three components on the determination of cloud fraction and the construction of the composite.

**Equation 2**
Is the comma correct here?

**p5, l14**
It is difficult to understand the correct domain of the gb-chromaticity diagram. What is exactly the (1/2, 1/2) point referring to?

**Section 3.3, p5, l26**
I don't understand why the functions max and min must ensure that cloud fraction is confined in the interval [0,1]. Aren't already the cloud free reflectances $\rho_{cf}$ the minimum available for the scene and aren't the $\beta$ already compensating for radiative affects? What are the physical units of the coefficients $\alpha$ and $\beta$? Are they unit-less?

**Section 3.3.1**
Recalling that specular reflection occurs when the viewing zenith angle equals the angle of illumination, given zero azimuth, could the authors briefly add an explanation of the need of a reflectance ratio criterium, instead of only geometrical consideration?

**Section 4, p6, l19-20**
It is said that the limitations of the CRB model are already noticeable with GOME-2. Where to find information on this? What limitations? Please, explain.

**p6, l22**

[Figure]

When the authors write that the layers are optically uniform, what properties are they addressing? LWP, droplet phase function, number concentration or? Please, add information on what optical properties are kept uniform.

**Section 4**

While the technique of wavelength recalibration is often omitted in modern papers about cloud remote sensing, it is relevant on its own. The authors might want to provide here more technical information so that the reader can independently implement it. Among the details to be provided, the following turn out to be useful: spectral sampling of the reference solar irradiance and source; fitting procedure, description of polynomials used in the spectral bins to find the optimal grid and iterations; value of calibration accuracy that can be achieved; references to past literature and technical documents, whenever appropriate (e.g., van Geffen and van Oss, 2003).

**Section 4.2, p7, l11**

How many scattering layers are clouds made of? Please, provide this information

**Section 4.4, p8, l12**

I am puzzled by the statement that the "desired total intensity I will incorporate the effects of polarization". Since we are placed in the NIR region and that the authors state that the thermodynamic phase of water is not relevant for the task under consideration (implying that the retrieval algorithm will not discriminate between water and ice, the latter best seen looking at Stokes Q), I do not see the strict need to simulate all components of the Stokes vector. Could you please clarify in the text how and why you do run VLIDORT? If you have pre-calculated all Stokes components, but you interpolate to find the match between measurement and forward intensity only for Stokes I? Is this a requirement for future applications at trace gas retrieval?

**p8, l15**

Please, provide the spectral resolution in nm instead of wavenumbers.

**p8, l21-22**
Please, state here whether your algorithm will be sensitive to the ice phase.

**p9, l9-13**

As far as I know, the accuracy of a neural network (NN) approach depends on the training set. Do I correctly understand that here the training set is purely synthetic and is made of NIR radiances, without external real datasets as, for instance, from measurements in the thermal infrared?

Moreover, I find confusing the role of the NN within the ROCINN framework for TROPOMI. In an earlier version of the ROCINN algorithm (Loyola et al., 2007), as applied to GOME measurements, the NN was used to solve the inverse problem, whereas the NN of this TROPOMI-ROCINN version solves the forward problem and the inversion is left to Tikhonov-Phillips.

If this is true, this information should be clearly stated in the paper to avoid confusion and justified from the perspective of the training sets. So, please, help the reader fully understand what development has been undertaken from the old ROCINN to this new version.

**Section 4.7**

This section has several shortcomings and seems to be written in haste. Basically, explanation of the results presented in all three figures and geophysical settings of this exercise are missing. I list my remarks in the following bullets.

1. The space of sampled geometries and cloud properties is not given. Thus, the reader does not know if the biases of the CRB retrieval (Figure 5) are coming from low-, mid- or high-level clouds.

2. Figure 4 is clearly not informative. Not only are the curves not color-coded, but one cannot understand what spectra are overlapping and why. I suggest to remove it, also because the shape of the oxygen A-band as function of changes of the main atmospheric properties under consideration is already well-known.

3. It is well-known that COT accuracy is strongly dependent on the viewing geometry. So, Figure 6 (left) should also address this information and provide the reader with more confidence that deviations from the 0-bias median are due to viewing-geometries (or are there other reasons?). Either increase the size bin of the x-axis, or color-code as function of VZA/SZA.

4. As long as the range of retrieved COT is not given, recalling that COT spans three orders of magnitude and that COT errors are usually non linear, the left plot of Figure 6 is little informative. So, please, provide more explanation on this aspect.

5. Figure 6 is not consistent, because COT bias is juxtaposed for one model (CAL) with the cloud albedo (CA) bias for the other model (CRB). And because no information is given on the correspondence between COT and CA, one cannot judge the performance of the two models within this task. So, either add also a CA bias plot for the CAL model and a COT bias plot for the CRB model or provide a clear description on why the two plots can be regarded as the manifestation of the same process/effect.

6. Please, define in text (and in the figures/captions) how are differences calculated. Are these relative or absolute errors?

7. Please, provide in the text a physical explanation why the cloud albedo difference is not symmetric about the 0-bias line, while the COT bias is, and why should CA be likely underestimated with the CRB model, as the red PDF is slightly skewed into the negative domain.

**Section 4.5, p10, l3**
What are the other options the inverse framework allows? If the narrative of the paper

requires this information, then provide it. Otherwise the sentence sounds odd and disconnected from the general flow.

**Section 5, p11, l20-21**
Could you provide exact figures on the error in COT due to uncertainties in surface albedo and size distribution parameters, in the same fashion you do for the influence of cloud geometrical fraction? The sentence is too general.

**p12, l 1-4**
Do you have a reference for the TROPOMI calibration exercise?

**Section 5.1, p12, l 9**
Where can the TROPOMI mapping tables be found? Are they publicly available? If yes, why not mention the source?

**Section 6**
It is clearly a matter of style, so, as suggestion, I would opt for compactness and avoid undue subsectioning, so that the flow of the paper isn't broken too much. I think it would suffices to rename the title of Section 6 and regroup the comparisons as follows

Section 6 "Application to OMI and GOME-2 and comparison with independent retrievals"
Section 6.1 "Comparison of OCRA with OMI and MODIS cloud fraction"
Section 6.2 "Comparison of ROCINN with GOME-2 cloud top height and thickness"

**Section 6.1, p13 l9**
I think the authors should check the sequence of figures, because the OCRA cloud-free background has numbering 2, while belonging to a later section.

**Section 6.1.1, p13, l23**
What kind of MODIS platform and product is? No reference is given here and the naming OMMYDCLD suggest that the authors use Aqua and not Terra. With this

respect, the different radiometric performance between Aqua and Terra could also impact the zonal comparison of Figure 8. But in absence of a clear reference, no judgment can be given.

**p13, l26-27**
Are the overpass times of OMI and MODIS comparable? Could you please add this information, if relevant for the differences found in the zonal plot?

**p13, l27**
Can the author substantiate with references or with a physical reasoning the statement "The UV sensors are not sensitive to optically thick clouds"?

**p14, l1-3**
While it is clear that fixing the albedo of a cloud at 0.8 (a too large value and to substantiate this statement you can cite Lelli et al, AMT 2012 - and report the mean global cloud albedo value of 0.63 and 0.55 from ROCINN) leads to a lower cloud fraction because the radiative balance within a pixel must be conserved (even if, strictly speaking, this general statement should be first checked against the RT assumptions of the respective cloud fraction algorithms), it is not clear why OMI-derived cloud fractions are still different from MODIS, even without assuming a fixed cloud albedo.

In absence of a quantitative and third cloud fraction source, it is not sound to say that OCRA and OMAERUV are underestimating (MODIS could overestimate as well), but still a physical explanation for this discrepancy should be given. Is this a geometrical, radiative or sampling effect? For the latter, I mention that if the L2 colocation procedure is avoided and the authors deploy a resampling of downstream daily gridded L3 to match OMI spatial resolution, then biases can occur. One should consider the number of available measurements with respect to the gradient of the cloud property within the spatial box to be gridded (cfr. Levy et al. 2009).

[Figure]

Figure 8 would be more informative if the zonal plots would be split for values above land and water masses.

**References**

Lelli L, Weber M and Burrows JP (2016) Evaluation of SCIAMACHY ESA/DLR Cloud Parameters Version 5.02 by Comparisons to Ground-Based and Other Satellite Data. Front. Environ. Sci. 4:43. doi: 10.3389/fenvs.2016.00043

Stengel, M., Stapelberg, S., Sus, O., Schlundt, C., Poulsen, C., Thomas, G., Christensen, M., Carbajal Henken, C., Preusker, R., Fischer, J., Devasthale, A., Willén, U., Karlsson, K.-G., McGarragh, G. R., Proud, S., Povey, A. C., Grainger, D. G., Meirink, J. F., Feofilov, A., Bennartz, R., Bojanowski, J., and Hollmann, R.: Cloud property datasets retrieved from AVHRR, MODIS, AATSR and MERIS in the framework of the Cloud_cci project, Earth Syst. Sci. Data Discuss., https://doi.org/10.5194/essd-2017-48, in review, 2017.

du Piesanie, A., Piters, A. J. M., Aben, I., Schrijver, H., Wang, P., and Noël, S.: Validation of two independent retrievals of SCIAMACHY water vapour columns using radiosonde data, Atmos. Meas. Tech., 6, 2925-2940, doi:10.5194/amt-6-2925-2013, 2013.

J. van Geffen and R. van Oss, Wavelength calibration of spectra measured by the Global Ozone Monitoring Experiment by use of a high-resolution reference spectrum, Appl. Opt. 42, 2739-2753 (2003).

Lelli, L., Kokhanovsky, A. A., Rozanov, V. V., Vountas, M., Sayer, A. M., and Burrows, J. P.: Seven years of global retrieval of cloud properties using space-borne data of GOME, Atmos. Meas. Tech., 5, 1551-1570, doi:10.5194/amt-5-1551-2012, 2012.

R. C. Levy, G. G. Leptoukh, R. Kahn, V. Zubko, A. Gopalan and L. A. Remer, A Critical Look at Deriving Monthly Aerosol Optical Depth From Satellite Data, in IEEE Transactions on Geoscience and Remote Sensing, vol. 47, no. 8, pp. 2942-2956, Aug. 2009.
doi: 10.1109/TGRS.2009.2013842

---

## Author Comment (AC1) · 7 Aug 2017

**Reply to Anonymous Referee #2**

Referee comments are written in black font.

Author replies are written in red font.

Changes in the revised manuscript are written in blue font.

Review of the manuscript by Loyola et al.

The manuscript describes the OCRA cloud fraction and ROCINN cloud pressure algorithms and their modifications that were made to adapt the algorithms for TROPOMI. Most material of the manuscript has been published. However, the paper contains some original material that is mainly related to the modification of the ROCINN algorithm. This material may be of interest for the developers of cloud algorithms for satellite hyperspectral radiometers. Moreover, the OCRA and ROCINN cloud algorithms were selected for TROPOMI; that is why it is important to document the algorithms in the literature. The paper subject is appropriate to AMT. Earlier work is recognized and

credited. The abstract provides a sufficiently complete summary of the paper. The paper is well organized. I think that the paper needs significant revisions before recommending it for publication. The authors should address the following comments.

**General comments**

1. The authors state that the cloud pressure algorithm, ROCINN_CAL, provides better cloud-top retrievals than ROCINN_CRB. The TROPOMI cloud products are intended to use in trace-gas retrievals. It is not obvious that the cloud-top pressures can produce better trace-gas retrievals.

The authors refer to the reply to "Section 1, Introduction, p 2, l 8-12" from referee #1.

The Mixed Lambertian Equivalent Reflectivity (MLER) model compensates for photon transport within a cloud by placing the Lambertian surface somewhere in the middle of the cloud instead of at the top. As clouds are vertically inhomogeneous, the pressure of this surface does not necessarily correspond to the geometrical center of the cloud, but rather to the so-called optical centroid pressure (OCP). Cloud OCPs are the appropriate quantity for use in trace-gas retrievals from satellite instruments. Cloud-top pressures are not equivalent to OCPs and do not provide good estimates of solar photon path lengths through clouds that are needed for trace-gas retrievals from ultraviolet and visible wavelength solar backscatter measurements (Ziemke et al., 2009; Joiner et al., 2012). The authors should prove that the ROCINN_CAL cloud-top pressures do produce better trace-gas (e.g. O3 or NO2) retrievals. That is particularly important in the view that the TROPOMI NO2 algorithm makes use of OCPs from the MLER-based FRESCO+ cloud algorithm (van Geffen et al., TROPOMI ATBD of the total and tropospheric NO2 data products, URL: https://sentinel.esa.int/web/sentinel/user-guides/sentinel-5p-tropomi/document-library, 2016).

J.R. Ziemke, J. Joiner, S. Chandra, P.K. Bhartia, A. Vasilkov, D.P. Haffner, K. Yang, M.R. Schoeberl, L. Froidevaux, and P.F. Levelt, Ozone mixing ratios inside tropical deep convective clouds from OMI satellite measurements, Atmos. Chem. Phys., 9, 573-583, 2009.

The authors emphasize that a full study of the impact on the accuracy of the trace gas retrieval is out of the scope of the present manuscript.

As already stated on page 15, lines 23-24, a forthcoming paper on the TROPOMI/S5P special issue will demonstrate that ozone total column accuracy is improved when using the CAL model. Furthermore, in section 2 of the paper we will add a summary and references to previous work

showing that cloud model is more appropriated than a Lambertian model for (a) the retrieval of aerosol properties from UV measurements (Torres, O., H. Jethva, and P. K. Bhartia, Retrieval of aerosol optical depth above clouds from OMI observations: Sensitivity analysis and case studies, J. Atmos. Sci., 69(3), 1037–1053, doi:10.1175/JAS-D-11-0130.1, 2011) and (b) the estimation of the surface UV irradiance (Krotkov, N. A., Bhartia, P. K., Herman, J. R., Ahmad, Z., and Fioletov, V.: Satellite estimation of spectral surface UV irradiance 2: Effect of horizontally homogeneous clouds and snow, J. Geophys. Res., 106, 11 743–11 759, 2001), moreover, this more realistic cloud model will be used for the surface UV products from TROPOMI (Lindfors, A. V., Kujanpää, J., Kalakoski, N., Heikkilä, A., Lakkala, K., Mielonen, T., Sneep, M., Krotkov, N. A., Arola, A., and Tamminen, J.: The TROPOMI surface UV algorithm, Atmos. Meas. Tech. Discuss., https://doi.org/10.5194/amt-2017-210, in review, 2017).

Finally please note that the same team that developed the MLER model published a paper showing that a plan-parallel cloud model is superior to a LER and MLER model for trace gas retrievals: "Although one of these models (MLER) can be adjusted to agree reasonably well with the TOMS data, the adjustments are somewhat arbitrary and may not be suitable for interpreting satellite data if one desires high accuracy." (Ahmad, Z., P. K. Bhartia, and N. Krotkov (2004), Spectral properties of backscattered UV radiation in cloudy atmospheres, J. Geophys. Res., 109, D01201, doi:10.1029/2003JD003395).

See also the comment and reply to "Section 1, Introduction, p 2, l 8-12" from referee #1.

Section 2 of the revised manuscript will be extended as described.

2. The OCRA algorithm has been described in detail in Loyola et al. (2007) and Lutz et al. (2016). In those papers, the authors used the normalized RGB (red-green-blue) representation of colors. In this manuscript the authors propose the Green-Blue color system for TROPOMI. This switching to the GB system should be explained because the red channel (675-775 nm) is available in TROPOMI. The authors should also compare cloud fraction retrievals from RGB and GB using e.g. GOME-2 data.

The switching from RGB to GB is mainly twofold: First, the TROPOMI UV/VIS and NIR footprints will have a spatial mis-alignment. Hence, the GB and R colors will not see the same ground pixel. And second, OMI which is needed to provide the cloud-free reflectance background maps, does not have channels in the red, which could be used to define a color R.

We shown with OMI data that the OCRA color space approach also works with two colors instead of three colors. Since a mis-alignment correction poses as an additional error source, it was decided to use the GB two color approach instead.

A comparison of OCRA cloud fraction retrievals for GOME-2 test data using RGB and GB only will be carried out and the results will be presented in the revised manuscript.

3. The use of the Mie scattering model of clouds is new in the ROCINN algorithm. That is why it is important to show that the selection of a single water cloud model, i.e. a single phase scattering function, is representative. Clouds can be multilayer and vertically-extended. This significantly affects photon path lengths and thus oxygen absorption in the cloud. The authors should show that their selection of a vertically uniform cloud model with a single geometrical thickness of 1 km is sufficiently representative. The authors should provide an estimate of possible cloud pressure errors associated with the selection of the cloud model.

The parameterization of single layer liquid water cloud is representative especially for low clouds (mean geometrical thickness approximately 1 km). In the oxygen A-band window, most of scattered radiation originates mainly from the cloud top because only a small portion of light penetrates into the cloud. Therefore, the selection of CGT of 1 km should be sufficient. In a previous study by Schuessler et al. (2014) the CTH retrieval was proven to be insensitive to the cloud geometrical thickness uncertainties. See also the author reply to the comment "p6, l22" of referee #1.

The authors will summarize the results from the sensitivity study quoted above in the revised manuscript. In order to tackle the uncertainties in the presence of multi-layer clouds, the authors will show the impact of double-layer clouds on the retrievals.

4. The authors should add a couple of paragraphs describing how their radiometric cloud fraction is used in the DOAS trace-gas algorithms. It is important to highlight the differences between the use of the radiometric cloud fraction and effective cloud fraction that comes from the MLER model.
The OCRA cloud fraction is being used in the operational DOAS trace-gas retrievals since GOME/ERS-2, a detail description on how OCRA cloud fraction is used in trace gas retrievals can be found in (Van Roozendael et al., 2006), (Valks et al., 2011) (Loyola et al., 2011). Furthermore, the usage of OCRA and ROCINN for the TROPOMI SO2 retrieval is described in (Theys et al., 2017).
This will be stated in the revised manuscript.

5. The authors show a comparison of the OCRA radiometric cloud fraction with the MODIS geometrical cloud fraction. It is unclear why the authors do not carry out a similar comparison of the ROCINN_CAL cloud-top pressure with the MODIS cloud-top pressure. This comparison should be done and quantitative results of the comparison should be provided.
OMI does not provide information on the oxygen A-band, which is why a ROCINN_CAL cloud-top pressure for OMI cannot be retrieved.

**Specific comments:**
Abstract. Some numbers characterizing the error budgets are strongly recommended in the abstract.
The error budgets for synthetic simulations and for GOME-2 measurements are given in section 4.7 and 6 respectively. Providing this information in the abstract will be misleading as the reader will be expecting the error budget for S5P but this can be assessed only when the S5P data become available.

Introduction. Please add the following reference and discuss how your approach differs from that by Diedenhoven et al. (2007). Diedenhoven et al., Retrieval of cloud parameters from satellite-based reflectance measurements in the ultraviolet and the oxygen A-band, JGR, 112, D15208, doi:10.1029/2006JD008155, 2007.
The authors will add the suggested reference. The authors acknowledge that the two approaches are similar in the sense that the three parameters CF, CTH and COT are retrieved and that both information from the UV and NIR are exploited. However, the authors emphasize that the two approaches are different in the following aspects: OCRA/ROCINN does not retrieve all three parameters simultaneously. It is a two step process, where OCRA first determines the CF from the UV/VIS region and then this CF is used as an a-priori input to ROCINN, which retrieves CTH and COT in the NIR.
Update the manuscript according to the points mentioned above.

P.4, L.13. "the minimum Lambertian equivalent" should be "the mixed Lambertian equivalent"
Indeed, it should say mixed instead of minimum.
The manuscript will be updated accordingly.

P.4, L.17. "in the range 330-390 nm" is incorrect; OMAERUV makes use of just two wavelengths 354 and 388 nm.
This is correct.
The manuscript will be updated accordingly.
P.5, L.27. Is it correct that the scaling and offset factors are determined using daily satellite measurements, not monthly?

The scaling and offset factors are based on histograms of the differences between measured reflectances and corresponding cloud free reflectances. The cloud free reflectances are based on *monthly* background maps derived as outlined in section 3.2. The histograms of the differences (rho – rho_CF), which are used to derive alpha and beta, are generated for *daily* global measurements, representing all possible cloud conditions. Several daily global histograms covering all seasons were generated in order to investigate the temporal evolution of these factors. Since no significant seasonal dependence was found, only one set of alphas and betas per color was fixed.
A short clarification will be added to the manuscript.

P.6, L.5. "a simplified sun-glint correction". Do you mean "sun-glint flagging"? Please provide information about the performance of the cloud algorithms over the sun glint area. For instance, this information can include the cross-track dependence of daily averaged OMI cloud fraction and cloud pressure for such areas.
The authors clarify that this is a flagging and not a correction. Please refer to answer in referee #1 comment on Section 3.3.1.
A short clarification will be added to the manuscript.

P.8, L.21-22. Your statement about small effect of the cloud phase (water or ice) should be proven by radiative transfer simulations. Please provide comparisons of computed TOA radiances for water and ice clouds and corresponding cloud pressure errors. Section 4.4. Please provide information about a number of computational nodes over surface reflectance, surface altitude, solar and viewing angles.
Mie theory is not sufficient to describe the scattering from ice crystals. Please see also the reply to comment "p8, l21-22" from referee #1.
The authors will reformulate the statement about the effect of the cloud phase.

Section 4.4. Please provide information about a number of computational nodes over surface reflectance, surface altitude, solar and viewing angles.
The node point generation, RTM simulation, and neural-network training has been done using the smart sampling and incremental function learning technique (Loyola et al., 2016). The input space (surface properties, cloud properties and geometry) in not sampled using a regular grid, but instead a technique which optimizes the distribution of multi-dimensional points within the (input) state space. The total number of computational nodes was of the order of some hundred thousands. The surface height and albedo were restricted between 0 to 4 km and 0 to 1, respectively. The CTH and COT were computed in the range 2-15 km and 2-50, respectively. The following geometry was covered: RAA in [0, 180$^o$], SZA in [0, 90$^o$] and VZA in [0, 75$^o$].
The node point generation is described in p.9, l.5-13. The total number of computational nodes will be added to the text.

P.9, L.11-12. Please provide typical errors of replacing exact radiative transfer simulations by neural network calculations for different sun-view geometries.
The mean average relative error over the O2 A-band spectral window for all scene geometries is below one percent.
This information will be included to the revised manuscript.

P.10, L.2-3. "the surface albedo climatology"; please provide a reference
The MERIS black-sky albedo climatology at 760 nm is used:
Popp, C., Wang, P., Brunner, D., Stammes, P., Zhou, Y., and Grzegorski, M., MERIS albedo climatology for FRESCO+ O2 A-band cloud retrieval, Atmos. Meas. Tech., 4, 463-483, 2011.
The above reference Popp et al., (2011) will be added to the manuscript.

P.10, L.4. "only very small changes are allowed" is a qualitative statement. Please provide quantitative information.

The very small changes here refer to the differences between the retrieved value of cloud fraction (and surface albedo) and their corresponding a priori value. The regularization parameter for cloud fraction and surface albedo is very high and thus, these parameters are always well within 1% difference from the a priori values.

This information will be added to the revised manuscript.

Section 4.6. describes well known theoretical estimates of the DFS and retrieval errors. Why the numerical estimates are not used in the text? I would remove this section and retain just a reference.

The authors prefer to keep the section.

At the end of Section 6.2.1., typical values for DFS and SIC will be added for the given GOME-2 test day (1$^{st}$ July 2012).

Section 5. Most statements in this section are qualitative like "can be accurately retrieved" (L.18), "quite sensitive to", "less significant are ROCINN errors" (L.20). The section titled "Error characterization" should provide quantitative information.

The authors agree to provide more quantitative information.

The section on error characterization will be updated in the revised manuscript.

P.11, L.19. Do you really mean "cloud geometrical fraction", not radiometric?

Correct.

The word geometrical will be removed.

P.12, L.4. Why the NIR stray light effects "will be assessed when the instrument provides measurements from space"? You say that "stray light issues were identified in the NIR band". The stray light contribution can be important for most absorption lines of the oxygen A-band. The authors should assess the stray light effects on the retrieved cloud properties. It seems to be straightforward to simulate stray light and investigate the impact on cloud pressure retrievals.

The authors agree to include the assessment of the stray light effect on retrievals.

The results of this assessment will be included in the updated version of the manuscript.

P.13, L.21-22. "This OMI cloud fraction is based on the filling-in of solar Fraunhofer lines caused by Raman scattering" is incorrect. The OMCLDRR cloud fraction is derived at 354 nm where the Raman scattering contribution is minimal.

Thank you for pointing this out. As stated in e.g. Joiner et al. (2012), the determination of the *cloud optical centroid pressure* "...makes use of the filling-in of Solar Fraunhofer lines by rotational-Raman scattering (RRS)…between 345 and 355nm...", whereas for the *effective cloud fraction* "a wavelength not significantly affected by RRS (354.1 nm)" is used.

In the revised manuscript, the sentence "This OMI cloud fraction is based on the filling-in of solar Fraunhofer lines caused by Raman scattering" will be replaced by "This OMI cloud fraction is derived at 354 nm where the contribution of Raman scattering is minimal".

P.13, L.27. Please explain why "the UV sensors are not sensitive to optically thick clouds". What physics do you mean in this statement?

This is a typo. It should say "thin" instead of "thick".

It will be corrected.

P.14, L.2-3. Please explain the meaning of "OCRA and OMAERUV report cloud fraction values more representative of the radiometric cloud fraction measured by the instrument". The instrument measures TOA radiances.

The referee is correct.

The authors will rephrase it as: "...representative of the radiometric cloud fraction based on the TOA radiances measured by the instrument".

---

## Author Comment (AC2) · 7 Aug 2017

**Reply to Anonymous Referee #1**

Referee comments are written in black font.
Author replies are written in red font.
Changes in the revised manuscript are written in blue font.

This paper accomplishes two tasks: first, it provides a scientific update of a cloud retrieval algorithm and, second, it summarizes the suite of cloud products that will be available upon the launch of the Sentinel-5 platform with the TROPOMI payload. As such, the readership can be wide and mainly made of two groups of individuals: experts in the field of cloud remote sensing and users of future TROPOMI data. This not only sets higher-than-usual requirements on the amount and quality of information to be conveyed in such a paper, but also demands a mixture of technical and scientific writing style.
In fact, while many concepts can be understood by the expert, users might not have the expertise and the required knowledge to understand the paper, especially when it goes down to error budgets and physical reasoning that support the conclusions of the presented work. Based on the above reasons, I think that this potentially important paper can be greatly improved with respect to readability and scientific information and publication should be warranted upon major revisions.

**Specific comments**
Abstract
The sentence provided on the higher accuracy of cloud properties derived from the NIR as compared to the TIR is indeed correct, but it is misleading for the reader, because she/he can think that a NIR-TIR comparison is one of the topic of the paper, which is not. So, this statement suits best as part of the introduction or the outlook, but not in the abstract, where, in my opinion, only an objective summary of the main matter of the paper should be given. As an interesting topic on its own (the NIR-TIR comparison), I flag to the authors that within Cloud_cci (Stengel et al., 2017), TIR retrievals from (A)ATSR are compared with retrievals derived from a combination of TIR and the NIR oxygen A-band channels by MERIS (Fig. 8, p. 21, third panel from above, CTP [hpa]). As outcome, one can appreciate that the addition of the oxygen A-band from MERIS corrects for photon penetration depth issues of the TIR channels and the found average bias amounts to approx 60 hpa, which translates to approx 0.8 km. Consistently, one recent study (Lelli et al. 2016) compared cloud properties derived from the oxygen A-band with the TIR-derived cloud heights of AATSR. It can be seen that TIR cloud retrievals are indeed placed lower (as the ROCINN_CRB) than the ones derived from the NIR with a scattering cloud layer model, as the ROCINN_CAL, by an average amount, again, in range 0.6 - 1.0 km. Once the accuracy of ROCINN_CAL and ROCINN_CRB will be assessed, it becomes reasonable to state that the oxygen A-band delivers more accurate cloud heights than the ones from TIR channels (albeit uncorrected).
The authors agree with the referee suggestion.
The sentence "Use of the oxygen A-band…" will be moved from the Abstract to the Introduction. The suggested references will be incorporated accordingly.

Section 1, Introduction, p 2, l 8-12
Keeping in mind, as research outlook, the impact that a change of the used cloud model in cloud retrieval algorithms can have on the accuracy of retrieved trace gas columns, I appreciate a more detailed presentation of past work (facts and figures) in the field. In fact, the sentence "These

studies have shown that cloud fraction..." is too general knowledge and does not properly convey the importance of the issue to be tackled.

It could be also somehow inaccurate, because when looking at du Piesanie et al (2013), the authors assessed the accuracy of SCIAMACHY water vapor columns as function of changing cloud fraction, optical thickness and cloud top height. They found that, using a scattering cloud model and the OCRA cloud fraction (making their results even more appropriate for this paper), CTH is the most critical parameter for water vapor, while cloud fraction and optical thickness are somewhat less relevant. So, please, expand this paragraph, briefly reporting past results about trace gas accuracy and information on the cloud model assumption that has been respectively used to derive them (wherever available and appropriate).

The authors agree with including previous studies carried out in the field. See also the reply to the first general comment of referee #2.

This part will be enhanced with more literature and with past results including the suggested reference.

Introduction, p2, l13

Is the spatial resolution the same for all TROPOMI bands? If not, please, report the correct information and briefly discuss how different footprint sizes can influence a joint exploitation (e.g., UV-Vis-NIR and SWIR).

The spatial resolution is not the same for all bands. Band 1 (UV) has 28x7 km$^2$ at nadir, bands 2-6 (UV, UVIS, NIR) have 3.5x7 km$^2$ at nadir and bands 7-8 (SWIR) have 7x7 km$^2$ at nadir.

In the revised manuscript, the spatial resolution will be given for each band and it will be emphasized that these footprints are specified for close-nadir viewing.

Introduction, p2, l17

Overpass time of the mentioned sensors? This is important for the extension of the data record, as different sensing times will record different atmospheres.

The authors agree that this is valuable information. The LST overpass times of the mentioned sensors are: GOME: 10:30 (D), SCIAMACHY: 10:00 (D), OMI: 13:30 (A), GOME-2A: 9:30 (D), GOME-2B: 8:45 (D), TROPOMI: 13:30 (A). Here, D and A denote Descending or Ascending, respectively.

This information will be included in the revised manuscript.

Section 2, p3, l5

What is OCRA CF needed for as "baseline input"?

The term "baseline" may be confusing.

The term "baseline" will be removed in the revised manuscript to read: "...from OCRA as an input."

p3, l10

It is said the the ROCINN_CAL is here presented for the first time. Then one might wonder where was the ROCINN_CRB model presented? Please, provide reference.

The latest versions of both ROCINN_CAL and ROCINN_CRB are presented in this manuscript.

The sentence will be re-phrased to: "...for the first time the latest developments of the ROCINN algorithm (incorporating both CAL and CRB models)."

p3, l10-17

This paragraph needs additional details on the errors as function of CRB/CAL, on the same line of thoughts of the impact of the cloud model on the accuracy of trace gases.

The authors agree.

This information will be included. Refer to the author response to the referee comment "Section 1, Introduction, p 2, l 8-12"

Section 2.1, p3, l24
References for ROCINN algorithm?
The authors agree.
References will be added in the revised manuscript.

p3, l27-28
Two aspects are not clear here. (1) why the IPA allows 1-D plane parallel RT of cloud-contaminated scenes and (2) whether the previous statement also holds for future TROPOMI measurements due to 3-D effects. Please, discuss this aspect.
This topic is already covered in Section 5, p11, l22-33.
A reference to Section 5 will be made in the revised manuscript.

p4, l6
PMD-derived cloud fraction benefits not only of the spectral coverage but also of a spatial resolution finer than the science channels. So, please, mention this.
The authors agree that this is valuable information.
The spatial resolution of the PMD footprints will be added in the revised manuscript.

p4, l10
The heritage OMI cloud fraction algorithms need a bit more details to make the reader understand how the cloud detection works. I might understand it, but it is not something all readers can follow.
The authors believe that an in-depth description of all further OMI cloud fraction algorithms is out of the scope of this manuscript. Several references are specified to guide the interested reader.
No change in the manuscript.

Section 3, p4, l22
Figure 1 contains a block which is not properly described in none of the following subsections, the "internal store". The authors need to address (and amend the manuscript where appropriate) the following questions:
(1) Why the need of a-priori selection if the brightness criterion should already deliver a minimum reflectance?
A-priori, because the cloud-free reflectance background needs to be known beforehand, usually based on a heritage instrument and then successively replaced by the target instrument as the mission goes on. Furthermore, the brightness criterion does not necessarily deliver a minimum reflectance. See also the reply to the comment for Section 3.3, p5, l26.
A short clarification will be added in the revised manuscript.

(2) What is the climatology used for?
The "climatology" are the cloud-free reflectance maps which give the rho_CF values in equation 3. Examples for these maps are given in Fig. 2.
A short clarification will be added in the revised manuscript.

(3) What climatology? Source, time-space aggregation? Quality of the values? Is a climatology appropriate and does it have shortcomings for the task?
The cloud-free reflectance background maps are based on OMI data from January 2005 to July 2008. They are generated separately for each color B and G. The temporal resolution is one month, i.e. for a given grid cell, all measurements from a given month are aggregated in order to reflect and cover seasonal surface variations. Finally, all data from January 2005, 2006, 2007 and 2008 are considered for the final map for January, etc. The spatial grid resolution is 0.2 degrees in latitude and 0.4 degrees in longitude. The cloud-free reflectance value for a given measurement is found via linear interpolation between the two adjacent monthly cloud-free maps. This approach of a linear interpolation between monthly maps was found to be the best tradeoff between the necessity to have

as many measurements as possible per grid cell in order to ensure a cloud-free situation among these (i.e. a long timescale is desired) and the necessity to be sensitive to rapid changes in the surface conditions like e.g. snowfall or melting (i.e. a short timescale is desired).
A short clarification will be added in the revised manuscript.

Section 3.1, p4, l28
It's the first time I read the terminology "ground-cover projection". What is this?
The authors agree that this is a confusing term.
The sentence will be changed to "...for the footprint of the measurement as:"

Section 3.2, p5, l8
It is said that reflectances are independent of atmosphere and line-of-sight. What do aerosol, Rayleigh and the surface do? Especially for the latter, does surface reflectivity change over the time needed to build the composite? This is crucial, especially when thinking at a small footprint. Please, add information on the impact of these three components on the determination of cloud fraction and the construction of the composite.
OCRA does not separate aerosols and clouds. The surface reflectivity change over the time is covered by the generation of monthly cloud-free reflectance maps with a linear interpolation between the two adjacent monthly cloud-free maps.
A short clarification will be added in the revised manuscript.

Equation 2
Is the comma correct here?
No it is not.
The second comma will be changed to a full stop.

p5, l14
It is difficult to understand the correct domain of the gb-chromaticity diagram. What is exactly the (1/2, 1/2) point referring to?
This point refers to a situation where B and G are equal, i.e. there is no wavelength dependence in the UV/VIS region which is interpreted as a scene fully covered by clouds. The OCRA assumption for a cloud is that the brightness is higher than the underlying surface (caution for snow/ice) and that the cloud spectrum is wavelength independent in the UV/VIS (i.e. B=G).
A short clarification will be added in the revised manuscript.

Section 3.3, p5, l26
I don't understand why the functions max and min must ensure that cloud fraction is confined in the interval [0,1]. Aren't already the cloud free reflectances ρcf the minimum available for the scene and aren't the β already compensating for radiative affects? What are the physical units of the coefficients α and β? Are they unit-less?
The cloud free reflectances rho_cf do not necessarily represent the minimum available for the scene. In the normalized gb-chromaticity diagram the situation furthest away from (½, ½) is searched and the corresponding B and G values of that individual measurement are written into the cloud free background map (i.e. the B and G values of the same grid cell belong to *one* individual measurement). Taking the absolute minimum B and G available would only work if B and G were treated independently. This is not done within OCRA.
The betas compensate for extremely "dark" scenes e.g. mountain shadows, solar eclipse tracks or partially for absorbing aerosols. The coefficients are only based on reflectance *differences* and hence unitless.
A short clarification will be added in the revised manuscript.

Section 3.3.1

Recalling that specular reflection occurs when the viewing zenith angle equals the angle of illumination, given zero azimuth, could the authors briefly add an explanation of the need of a reflectance ratio criterium, instead of only geometrical consideration?

The authors agree that this is a confusing paragraph.

The part with the reflectance ratios will be removed since it is irrelevant for the determination of the purely geometrical determination of the sunglint flag. On page 6, line 5, the sentence will be re-phrased to "…, a simplified sun-glint *flagging* will be used.".

Section 4, p6, l19-20

It is said that the limitations of the CRB model are already noticeable with GOME-2. Where to find information on this? What limitations? Please, explain.

The authors agree that the text should be expanded here.

Specific information on the limitations (i.e. overestimation of the ozone ghost column) and relevant references will be included in the revised manuscript.

p6, l22

When the authors write that the layers are optically uniform, what properties are they addressing? LWP, droplet phase function, number concentration or? Please, add information on what optical properties are kept uniform.

The cloud microphysical properties are not included in the current manuscript and the authors agree that this information needs to be specified. The authors consider only liquid water clouds (i.e. Cu, St, Sc) with a certain size distribution and a single phase scattering function in the parameterization. Revised manuscript will be updated including this information.

Section 4

While the technique of wavelength recalibration is often omitted in modern papers about cloud remote sensing, it is relevant on its own. The authors might want to provide here more technical information so that the reader can independently implement it. Among the details to be provided, the following turn out to be useful: spectral sampling of the reference solar irradiance and source; fitting procedure, description of polynomials used in the spectral bins to find the optimal grid and iterations; value of calibration accuracy that can be achieved; references to past literature and technical documents, whenever appropriate (e.g., van Geffen and van Oss, 2003).

Since this is not a technical paper, the authors propose to give only a high-level description of the recalibration procedure.

The following information will be added to the revised manuscript: source and sampling of solar reference, fitting procedure, description of polynomials. As Solar reference we use Chance and Kurucz, (2010).

Section 4.2, p7, l11

How many scattering layers are clouds made of? Please, provide this information

The authors consider only one scattering cloud layer.

The sentence will be re-phrased to "… cloud treated as a single scattering layer.".

Section 4.4, p8, l12

I am puzzled by the statement that the "desired total intensity I will incorporate the effects of polarization". Since we are placed in the NIR region and that the authors state that the thermodynamic phase of water is not relevant for the task under consideration (implying that the retrieval algorithm will not discriminate between water and ice, the latter best seen looking at Stokes Q), I do not see the strict need to simulate all components of the Stokes vector. Could you please clarify in the text how and why you do run VLIDORT? If you have pre-calculated all Stokes components, but you interpolate to find the match between measurement and forward intensity only for Stokes I? Is this a requirement for future applications at trace gas retrieval?

The reason of using the VLIDORT implementation including polarization instead of LIDORT is because a vector RTM is necessary for processing GOME, SCIAMACHY and GOME-2 data, which provide polarization information.
A short clarification will be added in the revised manuscript.

p8, l15
Please, provide the spectral resolution in nm instead of wavenumbers.
The authors agree.
The revised manuscript will be added accordingly.

p8, l21-22
Please, state here whether your algorithm will be sensitive to the ice phase.
The authors clarify that ROCINN is not sensitive to ice-phase clouds.
This part will be re-phrased in the revised manuscript.

p9, l9-13
As far as I know, the accuracy of a neural network (NN) approach depends on the training set. Do I correctly understand that here the training set is purely synthetic and is made of NIR radiances, without external real datasets as, for instance, from measurements in the thermal infrared?
The authors confirm that the training set is purely synthetic (VLIDORT simulations).
Moreover, I find confusing the role of the NN within the ROCINN framework for TROPOMI. In an earlier version of the ROCINN algorithm (Loyola et al., 2007), as applied to GOME measurements, the NN was used to solve the inverse problem, whereas the NN of this TROPOMI-ROCINN version solves the forward problem and the inversion is left to Tikhonov-Phillips. If this is true, this information should be clearly stated in the paper to avoid confusion and justified from the perspective of the training sets. So, please, help the reader fully understand what development has been undertaken from the old ROCINN to this new version.
The authors agree that a clarification needs to be added.
The information on the different usage of NNs in the previous and current algorithm versions will be included in the revised manuscript at the end of Section 4.

Section 4.7
This section has several shortcomings and seems to be written in haste. Basically, explanation of the results presented in all three figures and geophysical settings of this exercise are missing. I list my remarks in the following bullets.
1. The space of sampled geometries and cloud properties is not given. Thus, the reader does not know if the biases of the CRB retrieval (Figure 5) are coming from low-, mid- or high-level clouds.
The whole geometry space (only for VZA the range is from 0 to 75 degrees and not up to 90) is covered using the smart sampling technique. The CTH range was 2-15 km and the respective COT 2-50.
This information will be included in the revised manuscript

2. Figure 4 is clearly not informative. Not only are the curves not color-coded, but one cannot understand what spectra are overlapping and why. I suggest to remove it, also because the shape of the oxygen A-band as function of changes of the main atmospheric properties under consideration is already well-known.
The authors agree with the comment.
Figure 4 will be removed from the manuscript.

3. It is well-known that COT accuracy is strongly dependent on the viewing geometry. So, Figure 6 (left) should also address this information and provide the reader with more confidence that

deviations from the 0-bias median are due to viewing-geometries (or are there other reasons?). Either increase the size bin of the x-axis, or color-code as function of VZA/SZA.

The size bin of the x-axis can be increased.

Figure will be updated.

4. As long as the range of retrieved COT is not given, recalling that COT spans three orders of magnitude and that COT errors are usually non linear, the left plot of Figure 6 is little informative. So, please, provide more explanation on this aspect.

The retrieved COT varies from 2 to 50.

Information will be added to the figure.

5. Figure 6 is not consistent, because COT bias is juxstaposed for one model (CAL) with the cloud albedo (CA) bias for the other model (CRB). And because no information is given on the correspondence between COT and CA, one cannot judge the performance of the two models within this task. So, either add also a CA bias plot for the CAL model and a COT bias plot for the CRB model or provide a clear description on why the two plots can be regarded as the manifestation of the same process/effect.

It is not possible to retrieve CA from the CAL model or COT from the CRB model.

Information on the CA and COT relation will be added in the paper.

6. Please, define in text (and in the figures/captions) how are differences calculated. Are these relative or absolute errors?

These are absolute differences (a-b), and not relative differences.

This information will be mentioned in the caption and in the text.

7. Please, provide in the text a physical explanation why the cloud albedo difference is not symmetric about the 0-bias line, while the COT bias is, and why should CA be likely underestimated with the CRB model, as the red PDF is slightly skewed into the negative domain.

The CA difference is only slightly negative.

Explanation will be added in the text.

Section 4.5, p10, l3

What are the other options the inverse framework allows? If the narrative of the paper requires this information, then provide it. Otherwise the sentence sounds odd and disconnected from the general flow.

The authors agree that this formulation may cause confusion.

The text in the revised manuscript starting with ", but the inverse framework..." will be removed.

Section 5, p11, l20-21

Could you provide exact figures on the error in COT due to uncertainties in surface albedo and size distribution parameters, in the same fashion you do for the influence of cloud geometrical fraction? The sentence is too general.

The requested figures and information have been published in Schuessler et al., (2014).

The uncertainties in COT and the corresponding reference will be added to the revised manuscript.

p12, l 1-4

Do you have a reference for the TROPOMI calibration exercise?

The reference is the "NIR out-of-spectral band straylight analysis report" (S5P-KNMI-OCAL-0152-RP, issue 0.1, 2017-05-11, in review).

The above reference will be added to the manuscript.

Section 5.1, p12, l 9

Where can the TROPOMI mapping tables be found? Are they publicly available? If yes, why not mention the source?

The reference to the documentation of the mapping tables is "Sentinel 5 precursor interband coregistration mapping tables" (S5P-KNMI-L2-0129-TN, issue 4.0.0, 2015-11-23, released).

The reference to the documentation will be added to the revised manuscript.

Section 6

It is clearly a matter of style, so, as suggestion, I would opt for compactness and avoid undue subsectioning, so that the flow of the paper isn't broken too much. I think it would suffices to rename the title of Section 6 and regroup the comparisons as follows

Section 6 "Application to OMI and GOME-2 and comparison with independent retrievals"

Section 6.1 "Comparison of OCRA with OMI and MODIS cloud fraction"

Section 6.2 "Comparison of ROCINN with GOME-2 cloud top height and thickness"

The authors agree with the suggestion, the sub-subsections will be removed.

Section 6 will be re-structured according to the suggestions from the referee.

Section 6.1, p13, l8: Here is a typo in the manuscript. It must say from January 2005 to June 2008.

Section 6.1, p13 l9

I think the authors should check the sequence of figures, because the OCRA cloud-free background has numbering 2, while belonging to a later section.

The authors agree.

On page 5, line 18, the part "(see Figure 2 for example)" will be removed. Figure 2 will become Figure 7 and be introduced in Section 6.1.

Section 6.1.1, p13, l23

What kind of MODIS platform and product is? No reference is given here and the naming OMMYDCLD suggest that the authors use Aqua and not Terra. With this respect, the different radiometric performance between Aqua and Terra could also impact the zonal comparison of Figure 8. But in absence of a clear reference, no judgment can be given.

The used OMMYDCLD product provides the OMI/Aura and MODIS/Aqua merged cloud product. The proper reference is given in section 6.1.1 (third bullet point). No MODIS/Terra data are used.

It will be clarified in the manuscript that only Aqua, but no Terra data are used.

p13, l26-27

Are the overpass times of OMI and MODIS comparable? Could you please add this information, if relevant for the differences found in the zonal plot?

Since both Aura and Aqua are part of the A-train, the overpass times are comparable. The nominal separation between Aura and Aqua of 15 minutes was reduced to 8 minutes. The 8 minutes difference may become significant when comparing a single pixel during strong wind speeds, however for the averaging done for the zonal mean plots, the slightly different overpass times of OMI/Aura and MODIS/Aqua are not relevant and cannot be accounted for the shown differences.

It will be added to the revised manuscript that the overpass times of OMI and MODIS are comparable. The differences found in the zonal plot cannot be related to differences in overpass times.

p13, l27

Can the author substantiate with references or with a physical reasoning the statement "The UV sensors are not sensitive to optically thick clouds"?

This is a typo. It should say "thin" instead of "thick".

This will be corrected in the revised manuscript.

p14, l1-3

While it is clear that fixing the albedo of a cloud at 0.8 (a too large value and to substantiate this statement you can cite Lelli et al, AMT 2012 - and report the mean global cloud albedo value of 0.63 and 0.55 from ROCINN) leads to a lower cloud fraction because the radiative balance within a pixel must be conserved (even if, strictly speaking, this general statement should be first checked against the RT assumptions of the respective cloud fraction algorithms), it is not clear why OMI-derived cloud fractions are still different from MODIS, even without assuming a fixed cloud albedo. In absence of a quantitative and third cloud fraction source, it is not sound to say that OCRA and OMAERUV are underestimating (MODIS could overestimate as well), but still a physical explanation for this discrepancy should be given. Is this a geometrical, radiative or sampling effect?

The authors agree to add the information on mean global cloud albedo and to add the suggested reference. The authors emphasize that a direct comparison between the MODIS geometric cloud fraction and the OMI derived radiometric or effective cloud fractions should be treated with caution.

For the latter, I mention that if the L2 colocation procedure is avoided and the authors deploy a resampling of downstream daily gridded L3 to match OMI spatial resolution, then biases can occur. One should consider the number of available measurements with respect to the gradient of the cloud property within the spatial box to be gridded (cfr. Levy et al. 2009).

The authors clarify that OMMYDCLD product contains a MODIS cloud fraction already sampled to the OMI footprints.

Figure 8 would be more informative if the zonal plots would be split for values above land and water masses.

The authors agree to provide two separate figures (only land and only ocean) as suggested.

The manuscript will be updated with the points specified above.

References

Lelli L, Weber M and Burrows JP (2016) Evaluation of SCIAMACHY ESA/DLR Cloud Parameters Version 5.02 by Comparisons to Ground-Based and Other Satellite Data. Front. Environ. Sci. 4:43. doi: 10.3389/fenvs.2016.00043

Stengel, M., Stapelberg, S., Sus, O., Schlundt, C., Poulsen, C., Thomas, G., Chris-tensen, M., Carbajal Henken, C., Preusker, R., Fischer, J., Devasthale, A., Willén, U., Karlsson, K.-G., McGarragh, G. R., Proud, S., Povey, A. C., Grainger, D. G., Meirink, J. F., Feofilov, A., Bennartz, R., Bojanowski, J., and Hollmann, R.: Cloud property datasets retrieved from AVHRR, MODIS, AATSR and MERIS in the framework of the Cloud_cci project, Earth Syst. Sci. Data Discuss., https://doi.org/10.5194/essd-2017-48, in review, 2017.

du Piesanie, A., Piters, A. J. M., Aben, I., Schrijver, H., Wang, P., and Noël, S.: Vali-dation of two independent retrievals of SCIAMACHY water vapour columns using ra-diosonde data, Atmos. Meas. Tech., 6, 2925-2940, doi:10.5194/amt-6-2925-2013, 2013.

J. van Geffen and R. van Oss, Wavelength calibration of spectra measured by the Global Ozone Monitoring Experiment by use of a high-resolution reference spectrum, Appl. Opt. 42, 2739-2753 (2003).

Lelli, L., Kokhanovsky, A. A., Rozanov, V. V., Vountas, M., Sayer, A. M., and Burrows, J. P.: Seven years of global retrieval of cloud properties using space-borne data of GOME, Atmos. Meas. Tech., 5, 1551-1570, doi:10.5194/amt-5-1551-2012, 2012.

R. C. Levy, G. G. Leptoukh, R. Kahn, V. Zubko, A. Gopalan and L. A. Remer, A Critical Look at Deriving Monthly Aerosol Optical Depth From Satellite Data, in IEEE Transac-tions on Geoscience and Remote Sensing, vol. 47, no. 8, pp. 2942-2956, Aug. 2009. doi: 10.1109/TGRS.2009.2013842

---

## Author Response (AR2)

**Reply to Anonymous Referee #2**

Referee comments are written in black font.
Author replies are written in red font.
Changes in the revised manuscript are written in blue font.

Review of the manuscript by Loyola et al.
I think that the authors did not adequately address, at least, two comments which are important for evaluation of the algorithm error budget.

1. The authors state that the GB model was tested against the full RGB OCRA model and they found "only a small mean bias of 0.07" in the retrieved cloud fractions. No figure that illustrates the cloud fraction differences is provided. The cloud product is intended for the use in trace gas retrievals. For this purpose, the low cloud fraction range is most important. The mean bias of 0.07 is not small for low cloud fractions; it is quite large. The authors should compare cloud fractions from RGB and GB in more details and provide corresponding illustration. For example, they should look at the dependence of the mean bias on the RGB cloud fraction. The GB model may appear not to be appropriate for TROPOMI.
A new section 6.1 was included with the requested RGB and GB comparison.
The RGB/GB difference as a function of cloud fraction is shown in the new figure 8; for cloud fractions smaller than 0.1, the median and mean differences are 0.007 and 0.015 respectively.

Moreover, the red channel (675-775 nm) is available in TROPOMI; that is why the implementation of the RGB model has no problem for TROPOMI.
As explained in section 3 and section 5.1, TROPOMI UV/VIS and NIR channels are spatially misaligned. In other words, the GB and R components do not see the exact same footprint and therefore a RGB model cannot be applied to TROPOMI.

2. The authors removed the unsupported statement that "the use of water or ice properties has a relatively small impact on the O2 A-band spectral region". However, they ignored my suggestion to compare the simulated TOA radiances and provide an estimate of cloud pressure errors due to the use of a single scattering phase function for water clouds. The authors answered "Mie theory is not sufficient to describe the scattering from ice crystals". This is true. However the radiative transfer simulations do not necessarily require the Mie theory. The simulations do need a scattering phase function. Scattering phase function for ice clouds are available, e.g. at http://www.ssec.wisc.edu/~baum/Cirrus/IceCloudModels.html.
The requested simulations were performed. The retrieved cloud top height is almost unaffected (errors below 2%), and the errors on optical thickness are larger for optically thin clouds.
The results are summarized in section 5.

I recommend the manuscript for publication after the authors address these comments.
The remaining comments of the reviewer have been properly addressed in the updated paper.
We thank the reviewer for the suggestions which have helped to improve this manuscript.

The authors addressed most of the comments of the referees from the first round. There are a few questions still remaining.

On response to Referee #1
Comment on p.13, 127 The comment says "can the author substantiate with references or with a physical reasoning the statement 'The UV sensors are not sensitive to optically thick clouds' "? The response is that the statement should read "optically thin clouds". However, the authors still need to substantiate the revised statement as UV sensors should still be sensitive to thin clouds. Please quantify this statement as well.
We agree, the formulation in the paper was misleading and has been changed to:
The UV sensors are less sensitive to optically thin clouds than thermal infrared sensors

On response to Referee #3.
Comment 1. I think the authors missed the main point of the comment. The referee asks if their cloud model is more appropriate for use in the trace gas retrievals. That the author's model may be more appropriate for aerosol properties or surface UV is really irrelevant. I believe the main question is whether the cloud-top pressure is the quantity needed by the trace-gas algorithms (or is it cloud optical centroid pressure?). I don't think this question has been adequately addressed.
Note that trace gas retrievals require not only cloud pressure but also cloud optical thickness (cloud albedo).
The question of which cloud model is the most appropriate for trace gas retrievals has been addressed in previous publications, as already discussed in Section 2 of the paper: "Previous studies using TOMS and GOME/SCIAMACHY measurements have demonstrated that a plane-parallel scattering cloud model is superior to a Lambertian reflectance cloud model for trace gas retrievals (Ahmad et al., 2004) and (Diedenhoven et al., 2007) respectively".
We added the following paragraph to further emphasize the advantages of using the CAL model for the trace gas retrievals:
Furthermore, errors on retrieved $NO_2$ columns can be significantly reduced using cloud parameters from combined UV/VIS and NIR spectral (van Deelen et al., 2008) as obtained from OCRA and ROCINN_CAL.
Furthermore, a Lambertian cloud model requires additional ad-hoc implementations - ghost column and intra-cloud VCD corrections, see section 4 of the paper "The CRB approach was originally developed for GOME (footprint 320x40 km2), where different types of clouds are combined in the large satellite pixels and errors in the cloud model are compensated (Kokhanovsky et al., 2007), but the limitations of the CRB model are already noticeable with GOME-2 (footprint 80/40x40 km2) where an intra-cloud correction was developed to compensate the CRB overestimation of the O3 ghost column (Loyola et al., 2011)".

Finally the authors have evaluated the adequacy of 1-D versus 3-D cloud model for cloud and trace gas retrievals (Efremenko et al., 2016). The errors on clouds are already summarized in section 5, and the following text on the trace gas errors was added:

That paper also shows that the IPA leads to systematic errors in the retrieved ozone height-resolved partial columns

Comment 3: Just a general comment: While the Schuessler paper may have shown for some conditions that the CTH retrievals was "proven to be insensitive to the cloud geometrical thickness uncertainties", I do not believe this is true in general. What was shown was for a particular set of conditions that may or may be representative. Other papers, noted by the this Referee, do not substantiate this statement.

We do not claim that the retrieval is totally insensitive to cloud geometrical thickness. The exact wording in the paper is "The cloud retrievals are almost insensitive to cloud geometrical thickness uncertainties. In particular, for deviations of 50% in the cloud geometrical thickness, the retrieval errors in the cloud top height and cloud optical thickness are lower than 0.4 km and 2, respectively."

The Schuessler et al. paper used 25,986 simulations properly covering the geophysical conditions expected for a nadir viewing satellite with moderate spectral resolution. Other papers use far fewer simulations.

However, to avoid misunderstandings we added the following clarification in section 5:

[revised manuscript text omitted]